# Global simulation of dissolved $^{231}$Pa and $^{230}$Th in the ocean and the sedimentary $^{231}$Pa/$^{230}$Th ratios with the ocean general circulation model COCO ver4.0

Yusuke Sasaki[1], Hidetaka Kobayashi[1], and Akira Oka[1]

[1] Atmosphere and Ocean Research Institute, The University of Tokyo, Kashiwa, Japan

*Correspondence to*: Akira Oka (akira@aori.u-tokyo.ac.jp)

**Abstract.** Sedimentary $^{231}$Pa/$^{230}$Th ratios provide clues to estimate the strength of past ocean circulation. For its estimation, understanding the processes controlling the distributions of $^{231}$Pa and $^{230}$Th in the ocean is important. However, simulations of dissolved and particulate $^{231}$Pa and $^{230}$Th in the modern ocean, recently obtained from the GEOTRACES project, remain challenging. Here we report a model simulation of $^{231}$Pa and $^{230}$Th in the global ocean with COCO ver4.0. Starting from the basic water-column reversible scavenging model, we also introduced the bottom scavenging and the dependence of scavenging efficiency on particle concentration. As demonstrated in a previous study, the incorporation of bottom scavenging improves the simulated distribution of dissolved $^{231}$Pa and $^{230}$Th in the deep ocean, which has been overestimated in models not considering the bottom scavenging. We further demonstrate that introducing the dependence of scavenging efficiency on particle concentration results in a high concentration of dissolved $^{230}$Th in the Southern Ocean as observed in the GEOTRACES data. Our best simulation can well reproduce not only the oceanic distribution of $^{231}$Pa and $^{230}$Th but also the sedimentary $^{231}$Pa/$^{230}$Th ratios. Sensitivity analysis reveals that oceanic advection of $^{231}$Pa primarily determines sedimentary $^{231}$Pa/$^{230}$Th ratios. On the other hand, $^{230}$Th advection and bottom scavenging have an opposite effect to $^{231}$Pa advection on the sedimentary $^{231}$Pa/$^{230}$Th ratios, reducing their latitudinal contrast. Our best simulation shows the realistic residence times of $^{231}$Pa and $^{230}$Th, but simulation without bottom scavenging and dependence of scavenging efficiency on particle concentration significantly overestimates the residence times for both $^{231}$Pa and $^{230}$Th in spite of similar distribution of sedimentary $^{231}$Pa/$^{230}$Th ratios to our best simulation.

# 1 Introduction

The $^{231}Pa/^{230}Th$ ratios in marine sediments are used for estimating past ocean circulation strength (e.g., Yu et al., 1996; McManus et al., 2004; Gherardi et al., 2009; Böhm et al., 2015; Waelbroeck et al., 2018; Sufke et al., 2020). Alpha decay of $^{235}U$ and $^{234}U$ produces $^{231}Pa$ (half-life of ~32.5 kyr) and $^{230}Th$ (half-life of ~75.2 kyr), respectively, at an approximately constant $^{231}Pa/^{230}Th$ ratio of 0.093 in the ocean (Henderson and Anderson, 2003). $^{231}Pa$ and $^{230}Th$ are absorbed onto and desorbed from the surfaces of sinking particles (reversible scavenging; Bacon and Anderson, 1982) and eventually removed from the water column into marine sediments. Differential scavenging efficiencies of $^{231}Pa$ and $^{230}Th$ result in differences in their residence times in the ocean; the residence times of $^{231}Pa$ and $^{230}Th$ were estimated to be 111 and 26 years in Yu et al. (1996), and 130 and 20 years in Henderson and Anderson (2003). The shorter residence time of $^{230}Th$ indicates that $^{230}Th$ generated from $^{234}U$ is removed relatively quickly to marine sediments. On the other hand, the longer residence time of $^{231}Pa$ indicates that $^{231}Pa$ produced from $^{235}U$ is transported for a longer period by ocean transport. Therefore, the deviation of the sedimentary $^{231}Pa/^{230}Th$ ratios from the constant production ratio of 0.093 has been used as a proxy for ocean circulation (Yu et al., 1996). For example, the sedimentary $^{231}Pa/^{230}Th$ ratios from the Bermuda Rise were closer to 0.093 at the Last Glacial Maximum (LGM) than today, possibly suggesting that the Atlantic meridional overturning circulation (AMOC) was weaker at the LGM (McManus et al., 2004; Böhm et al., 2015). To use the sedimentary $^{231}Pa/^{230}Th$ ratios as a proxy for ocean circulation in a more quantitative manner, modeling about $^{231}Pa$ and $^{230}Th$ is important.

For $^{231}Pa$ and $^{230}Th$ modeling, one needs to take into account the different scavenging efficiencies of different marine particle types (e.g., particulate organic carbon, calcite, and opal) as well as the distribution of these particles (Chase et al., 2002; Edwards et al., 2005). Sinking particles effectively scavenge $^{231}Pa$ and $^{230}Th$ in regions with high particle concentrations. In general, $^{231}Pa$ has a longer residence time than $^{230}Th$, because sinking particles scavenge $^{230}Th$ more efficiently. However, as for opal particles, Chase et al. (2002) argue that opal scavenges $^{231}Pa$ more effectively than $^{230}Th$. This report is consistent with observational studies that find high $^{231}Pa/^{230}Th$ ratios in the Southern Ocean, where opal sinking flux is high (Rutgers van der Loeff and Berger, 1993; Walter et al., 1997; Chase et al., 2003).

Authors of previous modeling studies have tried to simulate the global distributions of $^{231}Pa$ and $^{230}Th$ by two-dimensional (2D) ocean models (Marchal et al., 2000; Luo et al., 2010) or three-dimensional (3D) ocean models of LSG-OGCM (Henderson et al., 1999), Bern 3D (Siddall et al., 2005; Rempfer et al., 2017), NEMO (Dutay et al., 2009; van Hulten et al., 2018), CESM (Gu and Liu, 2017) and iLOVECLIM (Missiaen et al., 2020a). There are also modeling studies that discuss the relationship between the strength of the AMOC and changes in sedimentary $^{231}Pa/^{230}Th$ ratios (Siddall et al., 2007; Lippold et al., 2012; Gu and Liu, 2017; Gu et al., 2020; Missiaen et al., 2020a; 2020b). Siddall et al. (2005) pioneered the 3D simulation of both $^{231}Pa$ and $^{230}Th$ by incorporating reversible scavenging. Their control simulation appropriately reproduced the observed distribution of sedimentary $^{231}Pa/^{230}Th$ ratios; it showed high sedimentary $^{231}Pa/^{230}Th$ ratios in regions where the sinking opal particle flux is high. In their control simulation, the concentrations of dissolved $^{231}Pa$ and $^{230}Th$ increased linearly with depth; this pattern agreed broadly with observed features. However, simulated dissolved $^{231}Pa$ and $^{230}Th$ were both higher than

observations in the deep ocean. In addition to reversible scavenging by sinking ocean particles, several studies (e.g., Anderson et al., 1983; Roy-Barman, 2009; Okubo et al., 2012) have pointed out the importance of additional scavenging at the seafloor (bottom scavenging) and the continental boundaries (boundary scavenging). The bottom scavenging has not been explicitly included in global 3D ocean models except for Rempfer et al. (2017) which used a simplified 3D ocean model of intermediate complexity similar to that used by Siddall et al. (2005) and reproduced the distributions of dissolved $^{231}$Pa and $^{230}$Th more realistically by introducing the bottom scavenging. On the other hand, Henderson et al. (1999) reproduced the distribution of dissolved $^{230}$Th in their ocean general circulation model (OGCM) simulation by changing the efficiency of reversible scavenging depending on particle concentration; this effect has not been directly considered by recent modeling studies but some studies have evaluated the impacts of changes in particle concentration and scavenging efficiency on the distribution of $^{231}$Pa and $^{230}$Th (van Hulten et al., 2018; Missiaen et al., 2020a and 2020b). Recently, the GEOTRACES project has led to a dramatic increase in the number of observations of dissolved and particulate $^{231}$Pa and $^{230}$Th (Schlitzer et al., 2018). The GEOTRACES database provides an opportunity to test models describing the cycling of these two radioisotopes in the global ocean. In this study, we report our model simulation about the global distribution of $^{231}$Pa and $^{230}$Th in the ocean with COCO ver4.0. Starting from the basic water-column reversible scavenging model of Siddall et al. (2005), we also introduced the bottom scavenging and the dependence of scavenging efficiency on particle concentration. Furthermore, we quantitatively discuss the processes that control the global distribution of sedimentary $^{231}$Pa/$^{230}$Th ratios; by performing a series of sensitivity simulations, we discuss how the individual processes (i.e., water-column reversible scavenging, ocean transport, and bottom scavenging) affect the global distribution of dissolved $^{231}$Pa and $^{230}$Th and sedimentary $^{231}$Pa/$^{230}$Th ratios.

## 2 Materials and Methods

### 2.1 Ocean general circulation model

The OGCM used in this study is COCO version 4.0 (Hasumi, 2006), the ocean component of the coupled ocean-atmosphere general circulation model MIROC version 3.2 (K-1 Model Developers, 2004). The COCO is also used as the ocean part of the MIROC earth system model (Hajima et al., 2020; Ohgaito et al., 2021). The model domain is global, with about one-degree horizontal resolution and 43 vertical layers. The vertical resolution varies from 5 (top) to 250 m (bottom). Surface boundary conditions are given from monthly averages of zonal and meridional components of wind stress, air temperature, specific humidity, net shortwave radiation, downward longwave radiation, freshwater flux, and wind speed. These boundary conditions are taken from the output of a pre-industrial simulation with the MIROC (Kobayashi et al., 2015; Oka et al., 2012). To calculate $^{231}$Pa and $^{230}$Th, we perform "offline" tracer simulation using physical fields obtained in advance by the COCO (Oka et al., 2008, 2009). The "offline" means that the calculation of tracer is separately performed from that of the physical field; since the distributions of $^{231}$Pa and $^{230}$Th do not affect the physical fields at all, the results do not depend on whether the model is "offline" or "online". The offline tracer model makes it easier to perform various sensitivity experiments. The tracer

model is integrated for 3,000 years and tracer fields reach a steady state where changes in ocean tracer inventory almost vanish (less than $10^{-5}\%$ per 100 years). We analyze the average of the last 100 years of the integration.

The physical fields used in this study is based on MIROC climate model simulations, and its reproducibility has been discussed and confirmed in a variety of literature (e.g., K-1 Model Developers, 2004; Gregory et al., 2005; Oka et al., 2006; Stouffer et al., 2006). We also note that the physical fields used here are the same as the pre-industrial (PI) simulation reported in Kobayashi et al. (2015) and Kobayashi and Oka (2018). For reference, the Atlantic meridional overturning circulation (AMOC) simulated by the COCO used in this study is shown in Fig. S11.

## 2.2 Particle fields

Following Siddall et al. (2005), the distribution of biogenic particles (organic carbon, calcite, and opal) is used to evaluate the scavenging of both $^{231}$Pa and $^{230}$Th. We define the concentration $M$ of each particle type [g m$^{-3}$] as $M = F/w_s$, where $F$ is the particle flux [g m$^{-2}$ yr$^{-1}$] and $w_s$ is the constant settling velocity [m yr$^{-1}$]. The particle flux is calculated using the export flux from the euphotic zone and an assumed vertical profile of each particle type. The detailed procedure is explained below.

First, the particulate organic carbon (POC) export flux from the euphotic zone is calculated by multiplying the distribution of primary production derived from satellite observations (Behrenfeld and Falkowski, 1997) by the export ratio (Dunne et al., 2005). From POC export flux and $M = F/w_s$, the concentration of POC at the base of the euphotic zone, $M_{POC}(z_0)$, where $z_0$ is the depth of the bottom of the euphotic zone, is obtained. After obtaining $M_{POC}(z_0)$, the POC concentration in the water column is expressed (Marchal et al., 1998) as

$$M_{POC} = M_{POC}(z_0) \left(\frac{z}{z_0}\right)^{-\varepsilon}, (1)$$

where $\varepsilon$ is a remineralization exponent for POC.

Next, the calcite and opal export fluxes from the euphotic zone are calculated by multiplying the POC export flux by their rain ratios, which are estimated following formulations of Siddall et al. (2005) and Maier-Reimer (1993); please refer to Eq. (2)–(5) of Siddall et al. (2005) for detail. The calcite particle concentration is calculated by assuming an exponentially decreasing vertical profile (Henderson et al., 1999; Marchal et al., 2000; Siddall et al., 2005). Thus, we have

$$M_{CaCO_3} = M_{CaCO_3}(z_0) \, exp \, exp \left(\frac{z_0 - z}{z_p}\right), (2)$$

where $z_p$ is the calcite penetration depth. While the opal concentration is expressed as an exponentially decreasing vertical profile in some previous studies (e.g., Henderson et al., 1999), we consider opal dissolution to be dependent on temperature, following Siddall et al. (2005), as

$$M_{opal} = M_{opal}(z_0) \, exp \, exp \left[\frac{D_{opal} \cdot (z_0 - z)}{w_s}\right], (3a)$$

$$D_{opal} = B(T - T_0), (3b)$$

where $D_{opal}$ $[yr^{-1}]$ is the opal dissolution rate, $T_0$ is the minimum temperature [°C] of seawater in the model, and $B$ is a dissolution constant $[°C^{-1} yr^{-1}]$. Table 1 lists the parameter values used in this study. Figure S10 shows the distribution of particle fluxes in the surface ocean.

## 2.3 Reversible scavenging model

We use a tracer model of $^{231}$Pa and $^{230}$Th based on Siddall et al. (2005). The dissolved concentration ($A_d$) and particle concentration ($A_p$) of $^{231}$Pa and $^{230}$Th are calculated from the following equations:

$$\frac{\partial A_{total}^i}{\partial t} = \beta^i - \lambda^i A_{total}^i - w_s \frac{\partial A_p^i}{\partial z} + Transport, \text{(4a)}$$

$$A_{total}^i = A_p^i + A_d^i. \text{(4b)}$$

In Eq. (4a), the first term on the right-hand side ($\beta^i$) represents production from uranium ($^{231}$Pa from $^{235}$U; $^{230}$Th from $^{234}$U), the second term represents radioactive decay, the third term represents the effect of vertical particle settling, and the fourth term represents ocean transport by advection and diffusion. The superscript $i$ represents the isotope type ($^{231}$Pa, $^{230}$Th).

By following a reversible scavenging model (Bacon and Anderson, 1982), the relationship between the radionuclide concentration in the dissolved phase ($A_d$) and particulate phase ($A_p$) is represented by the partition coefficient ($K_j^i$) as

$$A_p^i = \sum_j \quad A_{j,p}^i, \text{(5a)}$$

$$K_j^i = \frac{A_{j,p}^i}{A_d^i \cdot C_j}, \text{(5b)}$$

where subscript $j$ represents the particle type (organic carbon, calcite, opal) and $C_j$ is the dimensionless ratio of particle concentration to the density of seawater. The formulation of the reversible scavenging was also described in Oka et al., (2009, 2021) and readers can obtain its detailed description therein. The partition coefficient depends on the type of particles (Siddall et al., 2005). The partition coefficients of $^{231}$Pa and $^{230}$Th for each type of particle have been estimated in previous studies (Luo and Ku, 1999; Chase et al., 2002). Chase et al. (2002) show that opal scavenges $^{231}$Pa more efficiently than $^{230}$Th, whereas calcite scavenges $^{230}$Th more efficiently than $^{231}$Pa. Here we use partition coefficients following Chase and Anderson (2004), as in other previous modeling studies (Dutay et al., 2009; Gu and Liu, 2017; Siddall et al., 2005; Table 2). The model parameters are summarized in Table 1.

## 2.4 One-dimensional reversible scavenging model

In addition to the three-dimensional tracer model based on the OGCM, we use a simple, vertical, one-dimensional model, which was widely used in previous studies (e.g., Bacon and Anderson, 1982), to analyze simulation results in Section 4. In the one-dimensional model, we assume a steady state and ignore the effect of ocean transport in Eq. (4a). Furthermore, we do not take the radioactive decay term into account because it is much smaller than the production term. Under these assumptions, Eq. (4a) becomes

$$\beta^i - w_s \frac{\partial A_p^i}{\partial z} = 0. (6)$$

In this one-dimensional model, production by uranium radioactive decay (the first term on the left side of Eq. (6)) is balanced by vertical transport through particle settling (the second term on the left side of Eq. (6)). If we assume that $A_p^i$ is zero at the sea surface ($z = 0$), then Eq. (6) can be solved, leading to

$$A_p^i = \frac{\beta^i}{w_s} \cdot z. (7)$$

Equation (7) shows that the vertical profile of $A_p^i$ is determined from two parameters: $\beta^i$ and $w_s$. From Eq. (5), we have

$$A_p^i = \sum_j A_{j,p}^i = \left( K_{CaCO_3}^i \cdot C_{CaCO_3} + K_{opal}^i \cdot C_{opal} + K_{POC}^i \cdot C_{POC} \right) \cdot A_d^i$$

$$= \sum_j \ (K_j^i \cdot C_j) \cdot A_d^i. (8)$$

The dissolved concentration can be obtained from Eq. (7) and (8):

$$A_d^i = \frac{1}{\sum_j \ (K_j^i \cdot C_j)} A_p^i$$

$$= \frac{1}{\sum_j \ (K_j^i \cdot C_j)} \frac{\beta^i}{w_s} \cdot z. (9)$$

Equation (9) shows that the vertical profile of $A_d^i$ is determined by the particle settling speed, the partition coefficients, and the concentrations of each particle. By comparing results from the one-dimensional model and the three-dimensional tracer model, we can isolate the influence of ocean transport (i.e., advection, diffusion, and convection) on the simulated distributions of dissolved [231]Pa and [230]Th (see Section 4; Table 3).

## 2.5 Experimental design

This study conducts a series of OGCM experiments. First, we perform an experiment named Siddall_EXP using the same parameters and formulations as in Siddall et al. (2005). As stated in the Introduction, Siddall et al. (2005) was a pioneering 3D model for global simulation of both [231]Pa and [230]Th. This model is now a relatively old model and the reversible scavenging model introduced in this model is simpler than more recent models. However, this model appropriately reproduced the observed distribution of sedimentary [231]Pa/[230]Th ratios as shown in their Fig. 2 which appears not necessarily inferior to that in more recent models. Therefore, in this study, we start with Siddall_EXP where this most basic reversible scavenging model of Siddall et al. (2005) is introduced.

Second, we perform an experiment named BTM_EXP, in which we additionally take bottom scavenging into account. Following Rempfer et al. (2017), we simply set the deepest model grid layer as the nepheloid layer. The thickness of the nepheloid layer becomes equal to the thickness of the corresponding deepest model grid layer which varies between 5 and 250 m depending on the depth. The intensity of the bottom scavenging depends on two parameters: the partition coefficient ($K_{bottom}$) and the concentration ($C_{bottom}$) of the bottom particles. Our treatment about C$_{bottom}$ is the same that in Rempfer et

al. (2017); we assume a globally uniform value for C$_{bottom}$ ($6.0\times10^{-8}$ g cm$^{-3}$) which is within the range of $4.0\times10^{-8}$ to $1.65\times10^{-6}$ g cm$^{-3}$ observed in the benthic nepheloid layers in the North Atlantic (Lam et al., 2015). As for $K_{bottom}$, because our formulation of the reversible scavenging is not the same as Rempfer et al. (2017), we needed to find its appropriate parameter value. For this purpose, we perform a number of simulations with different bottom scavenging intensities by changing the value of $K_{bottom}$.

Third, we perform a sensitivity experiment named KREF_EXP concerned with the reference partition coefficient ($K_{ref}$). In KREF_EXP, in addition to varying the value of the partition coefficient for bottom particles ($K_{bottom}$), we also vary the values of the reference partition coefficients ($K_{ref}$) from those assumed in Siddall_EXP and BTM_EXP.

Finally, we perform an experiment named PCE_EXP, in which we incorporate the dependence of scavenging efficiency on particle concentration. In PCE_EXP, $K_{ref}$ is not assumed to be constant but varies according to the following formulation of Henderson et al. (1999):

$$K_{ref} = \left(\frac{C_{total}}{C_{ref}}\right)^{-0.42} \times 10^7, (10)$$

where $C_{total}$ [g cm$^{-3}$] is the total concentration of all sinking particles ($C_{total} = C_{CaCO_3} + C_{opal} + C_{POC}$) and $C_{ref}$ [g cm$^{-3}$] is the reference concentration. Note that the value of $C_{total}$ is differently specified on each grid whereas $C_{ref}$ is given as a globally uniform value. Due to the dependence of $K_{ref}$ on $C_{total}$, the scavenging efficiency becomes lower under higher particle concentrations and higher under lower particle concentrations. We conduct several simulations by varying $C_{ref}$ between $10^{-9}$ $and$ $10^{-6}$ $g$ $cm^{-3}$ (smaller $C_{ref}$ value leads to stronger scavenging). Although the observed decrease of the partition coefficient with increased bulk particle concentration is not entirely understood (Pavia et al., 2018), we will show that this particle concentration effect becomes important for controlling dissolved $^{230}$Th in some ocean regions.

## 3 Results

### 3.1 Dissolved $^{231}$Pa and $^{230}$Th along the Atlantic meridional transects

First, we discuss the results of Siddall_EXP, focusing on the meridional distribution of $^{231}$Pa and $^{230}$Th in the Atlantic Ocean. Figure 1 shows the dissolved concentrations of $^{231}$Pa and $^{230}$Th simulated in Siddall_EXP along the Atlantic 30°W transect, together with GEOTRACES data (see Fig. S1 for the location of observations referenced in this study). We confirm that the distributions of dissolved $^{231}$Pa and $^{230}$Th in Siddall_EXP are approximately the same as those reported in Siddall et al. (2005; their Fig. 2). Because $^{231}$Pa and $^{230}$Th exchange reversibly with sinking particles and are transported to the deep ocean, the dissolved $^{231}$Pa and $^{230}$Th concentrations increase with depth, both in the model simulation and in observations. However, as in Siddall et al. (2005), the model simulation overestimates dissolved $^{231}$Pa and $^{230}$Th concentrations at depths greater than 2,000 m and 1,000 m, respectively. For quantitative analysis, we perform a linear regression analysis between the simulation results and observed data from the GEOTRACES GA02 transect; we calculate the root mean square deviation

(RMSD), the correlation coefficient (R), and the slope of the linear regression (s) of modeled concentration versus measured concentration, as summarized in Table S1. The linear regression line slope indicates the model's ability to reproduce the observed distribution; it approaches 1.0 when the model simulation realistically reproduces the target distribution (Dutay et al., 2009; Gu and Liu, 2017). For Siddall_EXP, the slope of linear regression line is significantly larger than 1.0 for both $^{231}$Pa (s=1.88, R=0.72 and RMSD=0.15) and $^{230}$Th (s=4.44, R=0.89 and RMSD=1.31; Table S1). This overestimation in the deep ocean is also found in other previous model simulations (e.g., Dutay et al., 2009; Gu and Liu, 2017; van Hulten et al., 2018).

Next, to reduce the overestimation of the simulated concentrations in the deep ocean, we additionally incorporate bottom scavenging in benthic nepheloid layers (BTM_EXP). The dissolved $^{231}$Pa and $^{230}$Th distributions are shown in Fig. 2 and 3, respectively. As expected, the incorporation of bottom scavenging helps reduce $^{231}$Pa and $^{230}$Th concentrations in the deep ocean, improving the model's agreement with the data. As for the distribution of dissolved $^{231}$Pa, the model results come relatively close to the GEOTRACES data if $K^{Pa}_{bottom}$ is set to $5.0 \times 10^5$ (s=1.04, R=0.90 and RMSD=0.05; see CTRL_EXP in Table S1; Figs. 2c and 2d). This result confirmed the importance of the bottom scavenging, which was already reported from a previous global 3D model (Rempfer et al., 2017) and a regional eddy-permitting model (Lerner et al., 2020). On the other hand, it is difficult to reproduce the observed distribution of dissolved $^{230}$Th in BTM_EXP. With $K^{Th}_{bottom} = 1.0 \times 10^6$, the concentrations of $^{230}$Th in bottom waters come close to observed values (Figs. 3c and 3d), but the concentrations in the deep ocean (from 2000 m to 5000 m) remain overestimated. In the case of larger $K^{Th}_{bottom}$, the simulated $^{230}$Th concentrations approach observed values in the deep ocean but are significantly lower than measurements in bottom waters (e.g., $K^{Th}_{bottom} = 1.0 \times 10^7$ in Figs. 3g and 3h). These results indicate that modification of Siddall_EXP by considering bottom scavenging alone is not sufficient for accurately simulating $^{230}$Th distribution in our model. As shown in Rempfer et al. (2017) and Lerner et al. (2020), the appropriate selection of scavenging parameter coefficients is required for more realistic simulation. Because our reversible scavenging model (which is the same as Siddall et al., 2005; section 2.3) is not the same as Rempfer et al. (2017) and Lerner et al. (2020), we need to discuss the validity of a scavenging parameter coefficient in our model (i.e., $K_{ref}$). In the following experiments (i.e., KREF_EXP and PCE_EXP), we discuss more appropriate treatment about $K_{ref}$ by focusing solely on $^{230}$Th.

To reproduce the distribution of $^{230}$Th more realistically, we change the value of the reference partition coefficient ($K^{Th}_{ref}$) in addition to $K^{Th}_{bottom}$ (KREF_EXP). Figure 4 summarizes the results of KREF_EXP and shows the simulated vertical distributions of dissolved $^{230}$Th for various values of $K^{Th}_{ref}$ and $K^{Th}_{bottom}$ (see Fig. 4g). Note that, for example, the simulation R2_B5 means that $K^{Th}_{ref}$ is set to $2.0 \times 10^7$ and $K^{Th}_{bottom}$ to $5.0 \times 10^5$. In the cases where $K^{Th}_{bottom}$ is set to $5.0 \times 10^5$ (namely R2_B5, R4_B5, and R6_B5), the $^{230}$Th concentrations systematically change depending on $K^{Th}_{ref}$; as the reversible scavenging on sinking particles becomes stronger (i.e., for larger $K^{Th}_{ref}$), the concentrations of dissolved $^{230}$Th become smaller throughout the water column (Figs. 4c, 4e, and 4f). As discussed for BTM_EXP, it is also confirmed that the stronger bottom scavenging (i.e., larger $K^{Th}_{bottom}$) leads to the lower concentrations near the sea bottom (e.g., see R2_B5, R2_B10, and R2_B20). For

some combinations of water-column scavenging and bottom scavenging, simulations (e.g., R6_B5, R4_B5, R4_B10) reasonably reproduce the observed profile of dissolved $^{230}$Th concentration. Among our KREF_EXP simulations, the R6_B5 simulation (Fig. 4f) shows the slope of the linear regression line nearest to 1.0 (s=0.88, R=0.81, and RMSD=0.20; Table S1) where $K_{ref}^{Th}$ is higher ($K_{ref}^{Th} = 6.0 \times 10^7$) than for Siddall_EXP and BTM_EXP ($K_{ref}^{Th} = 1.0 \times 10^7$). In the R6_B5 simulation (Fig. 4f), the vertical profile of dissolved $^{230}$Th is significantly improved from that of Siddall_EXP (Fig. 1d) and BTM_EXP (Fig. 3). We confirmed that the R6_B5 simulation captures the observed features of the Atlantic transects of the GEOTRACES data (Fig. 5a). However, the R6_B5 simulation still underestimates the concentrations of dissolved $^{230}$Th from the surface to intermediate depths (see Fig. 4f). Also, the high concentrations of dissolved $^{230}$Th observed in the Southern Ocean in GEOTRACES data are not well reproduced (Fig. 5a). To address this issue, we performed additional simulations by slightly changing the values of $K_{ref}^{Th}$ and $K_{bottom}^{Th}$ from the R6_B5 simulation (not shown), but found that it is difficult to remove the above-mentioned deficiencies by merely changing the values of $K_{ref}^{Th}$ and $K_{bottom}^{Th}$ in KREF_EXP.

Finally, we discuss PCE_EXP, in which the dependence of scavenging efficiency on particle concentration is taken into account, according to Eq. (10). We conduct several simulations by varying the value of the reference concentration ($C_{ref}$) between $10^{-9}$ and $10^{-6}$ g cm$^{-3}$. Among these results, we here discuss the case with $C_{ref} = 10^{-7}$ g cm$^{-3}$, which shows the best agreement with observations. Compared to the case in which the dependence of scavenging efficiency on particle concentration is not considered (i.e., R6_B5 simulation of KREF_EXP), PCE_EXP is expected to show smaller (larger) $K_{ref}^{Th}$ for the higher (lower) concentration of sinking particles. In Fig. 5, we compare the simulated dissolved $^{230}$Th distribution obtained from PCE_EXP and R6_B5 simulation of KREF_EXP. Owing to the dependence of scavenging efficiency on particle concentration, PCE_EXP reproduces the vertical distribution of dissolved $^{230}$Th slightly better than KREF_EXP (Fig. 5d). The regression analysis also confirms that the agreement with the GEOTRACES data becomes improved in PCE_EXP (s=0.98 and R=0.84; CTRL_EXP in Table S1). It is worthy to note that the distribution in the Southern Ocean is significantly improved in PCE_EXP (Fig. 5b) compared to KREF_EXP (Fig. 5a) as a result of the non-uniform distribution of the reference partition coefficient $K_{ref}^{Th}$ (Fig. 5c). In the Southern Ocean, where particle concentration is relatively higher than in other regions (Honjo et al., 2008), the value of $K_{ref}^{Th}$ in PCE_EXP is lower than that in the R6_B5 simulation of KREF_EXP ($K_{ref}^{Th} = 6 \times 10^7$; i.e., $K_{ref}^{Th} \sim 7.8$) (Fig. 5c). Therefore, the concentration of dissolved $^{230}$Th in PCE_EXP becomes high compared to the KREF_EXP, which leads to a more realistic distribution of dissolved $^{230}$Th in the Southern Ocean. The distributions of $^{230}$Th simulated in previous modeling studies (e.g., Figs. 4 and 5 in Dutay et al., 2009; Fig. 2 in Siddall et al., 2005; Fig. 2 in Gu and Liu; Fig. 3 in Rempfer et al., 2017; Fig. 12 in van Hulten et al., 2018; Fig. S3 in Missiaen et al., 2020a) are basically similar to our result (Fig. 6b); however, our simulation is the best at reproducing the high concentration in the Southern Ocean. Hereafter, our best simulation (i.e., $K_{bottom}^{Pa} = 5.0 \times 10^5$ case of BTM_EXP for $^{231}$Pa and PCE_EXP for $^{230}$Th) is called CTRL_EXP (see Table 2 for parameter values of CTRL_EXP).

## 3.3 Particulate $^{231}$Pa and $^{230}$Th

By conducting a series of experiments described above, this study successfully reproduces the observed distributions of dissolved $^{231}$Pa and $^{230}$Th, shown again in Figs. 6a and 6b, respectively. In addition to dissolved $^{231}$Pa and $^{230}$Th, particulate $^{231}$Pa and $^{230}$Th simulated in CTRL_EXP are compared with the reported observations (Figs. 6c and 6d). In addition to the GEOTRACES dataset, we use several reported observations here (i.e., data referenced in Siddall et al., 2005, Marchal et al., 2007, and Lerner et al., 2020; namely, from Colley et al., 1995; Moran et al., 1997; Moran et al., 2001; Rutgers van der Loeff and Berger, 1993; Vogler et al., 1998; Walter et al., 1997; Cochran et al., 1987; Moran et al., 2002; Guo et al., 1995). The model captures the observed tendency that the concentration becomes higher in the high-latitude Southern Ocean, as reported in previous studies (e.g., see Fig. 2 in Siddall et al., 2005). The ratio of $^{231}$Pa to $^{230}$Th in the particulate phase in the water column shows low concentrations in the deep ocean, while the ratio becomes high in the northern North Atlantic Ocean and the Southern Ocean (Fig. 6f). This feature is consistent with observational findings and recent modeling studies (e.g., Fig. 2 in Gu and Liu 2017; Fig. 3 in Rempfer et al., 2017). Although the number of available observations is limited for the particulate phase, it is confirmed that our simulation reasonably reproduces observed distributions for both dissolved and particulate phases.

## 3.4 Sedimentary $^{231}$Pa/$^{230}$Th ratios

Our CTRL_EXP also well reproduces the global distribution of sedimentary $^{231}$Pa/$^{230}$Th ratios (Fig. 6e) compared with the reported observations (Mangini & Sonntag, 1977; Muller & Mangini, 1980; Anderson et al., 1983; Shimmield et al., 1986; Schmitz et al., 1986; Yang et al., 1986; Shimmield & Price, 1988; Yong Lao et al., 1992; François et al., 1993; Frank et al., 1994; Frank, 1996; Bradtmiller et al., 2014, Luo et al., 2010, and their supplemental data). Sedimentary $^{231}$Pa/$^{230}$Th ratios are high along the margin of the North Pacific and the North Atlantic, as well as in the Southern Ocean, where particle concentrations are high. On the other hand, sedimentary $^{231}$Pa/$^{230}$Th ratios are low in the low-latitude regions, including subtropical gyres, where particle concentrations are low. These simulated features are consistent with observations (circles in Fig. 6e). Previous modeling studies reported the similar distribution of sedimentary $^{231}$Pa/$^{230}$Th ratios (e.g., Fig. 2 in Siddall et al., 2005; Fig. 11 in Dutay et al., 2009; Fig. 4 in Gu and Liu, 2017; Fig. 10 in Hulten et al., 2018; Fig. 4 in Missiaen et al., 2020) and our Siddall_EXP also reasonably reproduced the global distribution of sedimentary $^{231}$Pa/$^{230}$Th ratios (Fig. S4a). However, as shown above, the distributions of dissolved $^{231}$Pa and $^{230}$Th in the ocean are significantly different between CTRL_EXP and Siddall_EXP. Thus, each experiment implies a different set of processes controlling the distribution of sedimentary $^{231}$Pa/$^{230}$Th ratios. We will discuss this point later in the next section.

# 4 Discussion

## 4.1 Comparison with previous modeling studies

We demonstrated that our CTRL_EXP can reproduce a more realistic distribution of dissolved $^{231}$Pa and $^{230}$Th along the Atlantic meridional transects than Siddall_EXP by considering the bottom scavenging and the dependence of scavenging efficiency on particle concentration. Here, we compared our results with previous modeling studies which showed their model results along with Atlantic meridional transects (GEOTRACES GA02 section).

As far as we know, Rempfer et al. (2017) was the only 3D global ocean model which introduces the bottom scavenging, and in our study, we introduced the bottom scavenging into the global OGCM for the first time. Models without the bottom scavenging tend to overestimate the dissolved $^{231}$Pa and $^{230}$Th in the deep ocean as in our Siddall_EXP. For example, in Gu and Liu (2017) in which $^{231}$Pa and $^{230}$Th tracer are introduced into CESM1.3, their simulated $^{231}$Pa and $^{230}$Th concentrations are significantly overestimated in the deep ocean along the GEOTRACES GA02 section (their Fig. 2). In Dutay et al. (2009) in which $^{231}$Pa and $^{230}$Th tracer are introduced into NEMO-PISCES, influences of particle size and type on $^{231}$Pa and $^{230}$Th are discussed by performing several sensitivity simulations but all of their simulations overestimate $^{231}$Pa and $^{230}$Th concentrations in the deep Atlantic Ocean (their Figs 4 and 5, respectively). In van Hulten et al. (2018) which was the updated $^{231}$Pa and $^{230}$Th simulation with NEMO-PISCES, the model still overestimates $^{231}$Pa and $^{230}$Th concentrations in the deep Atlantic Ocean (their Fig. 12) because particles in the nepheloid layers (i.e., bottom scavenging) are not included in their model.

Although the incorporation of bottom scavenging is important for controlling the scavenging efficiency, it is worthy to note that bottom scavenging is not the sole process that controls the scavenging efficiency. Therefore, for example, the model which specified the relatively stronger affinity to the particle can lead to smaller tracer concentration even if the model does not include the bottom scavenging. In fact, in Missiaen et al. (2020), their simulated $^{231}$Pa and $^{230}$Th are underestimated in both the upper and deep oceans (their Supplementary Fig. 3) even if the bottom scavenging was not included in their model. This is because their specified scavenging parameters are relatively stronger than the other models (see their Table 2) as a result of their parameter tuning without the bottom scavenging.

In the Re3d_Bt_Bd simulation reported in Rempfer et al. (2017) where the bottom scavenging process is considered, the above-mentioned overestimation in the deep ocean was relaxed and their simulated distribution appears similar to our CTRL_EXP. Their study is the first 3D model demonstration about the importance of the bottom scavenging process, which is confirmed again in our study (e.g., from comparison with Siddall_EXP and KREF_EXP). However, their model still tends to somewhat overestimate the dissolved $^{231}$Pa compared with GEOTRACES GA02 data (their Fig.3). Because the formulation of the reversible scavenging and their model parameters are not the same as our CTRL_EXP, we expect that different choice of model parameter values leads to such differences; more specifically, our choice of $K_{ref}^{Pa}$ is based on Chase et al. (2002) and Siddall et al. (2005) whereas the scavenging efficiency parameters in Rempfer et al. (2017) are similar to those in Luo et al. (2010) and Marshall et al. (2002). In addition, as for $^{230}$Th, the high concentration in the Southern Ocean is not reproduced in their model, whereas this is reproduced in our CTRL_EXP by considering the dependence of scavenging efficiency on particle

concentration. Although the dependence of scavenging efficiency on particle concentration was already introduced in Henderson et al. (1995), our study demonstrates its importance for reproducing high concentrations in the Southern Ocean reported in GEOTRACES GA02 data for the first time.

This study newly introduces a $^{231}$Pa/$^{230}$Th model to the existing global three-dimensional OGCM. Based on the reversible scavenging model, this study well reproduces the distribution of dissolved concentration of $^{231}$Pa and $^{230}$Th by considering the bottom scavenging and the dependence of the scavenging efficiency on particle concentration. The importance of bottom scavenging on the dissolved concentration of $^{231}$Pa and $^{230}$Th is already discussed in previous studies (Rempfer et al., 2017; Lerner et al., 2020). Therefore, our result should be viewed as a confirmation of these previous results. However, we emphasize that this study provides a new estimate of the contribution of bottom scavenging to the distribution of sedimentary $^{231}$Pa/$^{230}$Th ratios compared to other processes such as advection and water-column scavenging. Rempfer et al. (2017) evaluated the performance of their $^{231}$Pa and $^{230}$Th simulations based on the root mean squared deviation normalized by the standard deviation of observations. In our control experiment (CTRL_EXP), the RMSD between the available GEOTRACES data is 0.57 for dissolved $^{231}$Pa and 0.51 for dissolved $^{230}$Th. These values lie in the range of values for the "standard" and "optimal" experiments of Rempfer et al. (2017), the latter of which considers both bottom scavenging and boundary scavenging (see Fig. 5 in Rempfer et al., 2017). Lerner et al. (2020) use a regional eddy-permitting ocean circulation model and focus on the western North Atlantic. They also point out that removal in the nepheloid layer significantly impacts the basin-scale distribution of dissolved and particulate phases of $^{231}$Pa and $^{230}$Th. In line with these previous studies, our result confirmed the importance of boundary scavenging. Recently, Gardner et al. (2018) reported data on the distribution of particles in benthic nepheloid layers. If such datasets become available for specifying the global distribution of particles in nepheloid layers, the effect of bottom scavenging can be introduced more realistically. It is also expected that additional consideration about boundary scavenging helps to improve our model simulation.

In addition to the bottom scavenging, our study highlights the importance of the dependence of scavenging efficiency on particle concentration. Although the decrease of the partition coefficient with increased bulk particle concentration has been reported from observations, the dependence of scavenging efficiency on particle concentration considered in PCE_EXP is not entirely understood (Honeyman et al., 1988; Henderson et al., 1999; Hayes et al., 2015). Recently, the particle concentration effect on $^{231}$Pa and $^{230}$Th partition coefficients in the open ocean along the GEOTRACES GA03 transect has been reported (Hayes et al., 2015; Lerner et al., 2017). Their study suggests that the dependency in the open ocean may deviate from Eq. (10). In discussing the factors responsible for the particle concentration effect, Pavia et al. (2018) point out the possibility that the particle concentration effect is an artifact caused by filtration. Further research is needed to elucidate the mechanisms that control the particle concentration effect.

As pointed out in previous studies (Rempfer et al., 2017; Lerner et al., 2020), the distributions of particle phases of $^{231}$Pa and $^{230}$Th are difficult to be reproduced in the model compared with the dissolved phases. For dissolved $^{230}$Th, the correlation coefficient between their model and observations is 0.80 for the GEOTRACES GA02 transect and 0.78 for the GA03 transect, comparable to our CTRL_EXP of 0.84 and 0.70, respectively. Part of the error could be related to the particle

fluxes that we give as an empirical distribution based on satellite observations. A $^{231}$Pa/$^{230}$Th modeling study using an ecosystem model that considers six different particles well reproduces the distribution of $^{231}$Pa and $^{230}$Th with a simple reversible scavenging model (van Hulten et al., 2018). More detail treatment of particles might be helpful for more realistic simulation of particulate $^{231}$Pa and $^{230}$Th. Furthermore, by examining the response of $^{231}$Pa and $^{230}$Th to freshwater forcing into the North Atlantic, Missiaen et al. (2020b) show that changes in biogenic particle fluxes may have caused 30% of the changes in the sedimentary $^{231}$Pa/$^{230}$Th ratios during the Heinrich stadial 1. Also, in Gu & Liu, (2017), the particle change due to freshwater and its impact on sedimentary $^{231}$Pa/$^{230}$Th ratios is examined. Therefore, the role of particle fields on the distribution of $^{231}$Pa and $^{230}$Th, which was not directly investigated in this study, needs to be further discussed in a future study.

## 4.2 Reproducibility along GEOTRACES GA03 and GP16 transects

So far, we have compared our model results with observations by focusing on the Atlantic meridional GEOTRACES transects (i.e., GA02 and GIPY05). Here, we will compare our CTRL_EXP with other available GEOTRACES transects: GA03 in the subtropical North Atlantic (Hayes et al., 2015) and GP16 in the South Pacific (Pavia et al., 2018).

Figure S5 shows the results of CTRL_EXP along with the GEOTRACES GA03 data. For dissolved $^{231}$Pa, the model shows a high concentration around a depth of about 3000 m and higher concentrations on the eastern/southern side of the basin as in observations (Fig. S5a). This feature was also well reproduced in the Re3d_Bt_Bd simulation of Rempfer et al. (2017) as shown in their Figure 2, but not in other previous models (e.g., Fig. 8 in van Hulten et al., 2018; Fig. 3 in Gu and Li, 2017). This confirms that the consideration of the bottom scavenging is helpful for improving the model result along the GEOTRACES GA03 section. For dissolved $^{230}$Th, features similar to $^{231}$Pa are also found in both the model and observations although the model appears to underestimate north-south or western-eastern differences (Fig. S5b). For particulate $^{231}$Pa and $^{230}$Th, the model tends to simulate high concentration near the sea bottom and the continental margins where the particle concentration becomes high, but such features are not necessarily clear in the GEOTRACES data (Figs. S5c and S5d). Our model may not sufficiently reproduce the bottom and boundary scavenging associated with terrestrial particles in this region. More sophisticated treatment of bottom and boundary scavenging might be required for addressing these issues.

Figure S6 shows the results of CTRL_EXP along with the GEOTRACES GP16 data. As with the other section data, CTRL_EXP approximately reproduces the distribution of $^{231}$Pa and $^{230}$Th. The observational data shows a clear signal associated with hydrothermal vents: low concentrations of dissolved $^{231}$Pa and $^{230}$Th and high concentrations of particulate $^{231}$Pa and $^{230}$Th, which are not simulated in our model. It has been pointed out that trace metals from hydrothermal activities may cause additional removal of $^{231}$Pa and $^{230}$Th (Shimmield and Price 1988; Lopez et al., 2015; Rutgers van der Loeff et al., 2016; German et al., 2016). Along the GEOTRACES GP16 section, $^{231}$Pa and $^{230}$Th have been found to decrease with increasing trace metals of iron and manganese supplied from hydrothermal vents (Pavia et al., 2018). Processes related to the hydrothermal vents are not explicitly incorporated in the present $^{231}$Pa and $^{230}$Th model simulations; its detailed treatment is beyond the scope of this study but appears necessary for more realistic simulations.

## 4.3 Processes controlling sedimentary $^{231}$Pa/$^{230}$Th ratios

In this subsection, we discuss the processes controlling the global distribution of sedimentary $^{231}$Pa/$^{230}$Th ratios. For this purpose, we decompose the processes controlling sedimentary $^{231}$Pa/$^{230}$Th ratios simulated in our best simulation CTRL_EXP into three parts: water-column reversible scavenging, three-dimensional ocean transport, and bottom scavenging. To evaluate how these three processes affect the distribution of $^{231}$Pa/$^{230}$Th ratios, we conduct two additional experiments (see Table 3). The first experiment is 3D_EXP, which is the same as CTRL_EXP except that bottom scavenging is not taken into account (i.e., we set $K_{bottom}^{Pa} = K_{bottom}^{Th} = 0$ in 3D_EXP). The second is 1D_EXP, which is the one-dimensional reversible scavenging model experiment described in Section 2.4. The tracer distribution in 1D_EXP is determined solely by the one-dimensional vertical process of reversible scavenging; the strength of scavenging changes spatially through changes in the partition coefficient ($K_j^i$ of Eq. (9) in Section 2.4) that depends on the specified three-dimensional particle concentration ($C_j$ of Eq. (9)). By using results of CTRL_EXP, 3D_EXP, and 1D_EXP, we can extract the influence of three processes: the influence of the one-dimensional vertical reversible scavenging is revealed by 1D_EXP, the influence of bottom scavenging is revealed by the difference between CTRL_EXP and 3D_EXP, and the influence of ocean transport is revealed by the difference between 3D_EXP and 1D_EXP. When we focus on sedimentary $^{231}$Pa/$^{230}$Th ratios, each process described above can be further examined for $^{231}$Pa and $^{230}$Th individually. For example, the difference in $^{231}$Pa/$^{230}$Th ratios between CTRL_EXP and 3D_EXP represents the influence of bottom scavenging of both $^{231}$Pa and $^{230}$Th, whereas the influence of bottom scavenging of $^{231}$Pa alone can also be evaluated from CTRL_EXP and 3D_EXP (i.e., $^{231}$Pa(CTRL)/$^{230}$Th(3D) minus $^{231}$Pa(3D)/$^{230}$Th(3D)).

In 1D_EXP, the particulate concentration is obtained from Eq. (7); the particulate concentration increases linearly with depth (Figs. S2c and S2d). The dissolved concentration is calculated from Eq. (9), suggesting that the concentration becomes higher for a lower partition coefficient ($K_j^i$ in Eq. (9)) and for a lower particle concentration ($C_j$ in Eq. (9)). Mainly due to the dependency on $C_j$, the dissolved concentration becomes higher (lower) in the area with lower (higher) particle concentration in 1D_EXP. As a result, the dissolved concentration becomes very high in the deeper ocean, where the particle concentration becomes lower for both $^{231}$Pa and $^{230}$Th (Figs. S2a and S2b). It is interesting to point out that the spatial pattern of dissolved $^{231}$Pa and $^{230}$Th (Figs. S2a and S2b) is similar to that of $K_{ref}$ in PCE_EXP (Fig. 5c) because both are affected by the amount of particle concentration. More importantly, although it is well known from previous studies, we emphasize here that the sedimentary $^{231}$Pa/$^{230}$Th ratios in 1D_EXP become uniform everywhere (0.093; Fig. S2e) because, as confirmed from Eq. (7), the ratio of particulate $^{231}$Pa to particulate $^{230}$Th amounts everywhere to $\beta^{Pa}/\beta^{Th} = 0.093$, regardless of geographic location (Fig. S2f).

In 3D_EXP, three-dimensional ocean transport operates, in addition to water-column scavenging considered in 1D_EXP (Fig. S3). As described above, the influence of ocean transport can be evaluated from the difference between 3D_EXP and 1D_EXP (Fig. 7). On the other hand, the influence of bottom scavenging can be obtained from the difference between CTRL_EXP and 3D_EXP (Fig. 8). Note again that since the sedimentary $^{231}$Pa/$^{230}$Th ratios in 1D_EXP are globally uniform

($^{231}$Pa/$^{230}$Th = 0.093), their spatial distribution is controlled not by the one-dimensional vertical process but by the ocean transport. Figures 7e and 7f demonstrate that the ocean transport effect captures the overall features of CTRL_EXP (Figs. 6e and 6f). On the other hand, bottom scavenging tends to cancel the effects of ocean transport and weaken the spatial contrast of $^{231}$Pa/$^{230}$Th ratios simulated in CTRL_EXP (Figs. 8e and 8f).

To evaluate the above processes controlling the sedimentary $^{231}$Pa/$^{230}$Th ratios in more detail, we further decompose the ocean transport contribution into those from $^{231}$Pa and $^{230}$Th, separately (Fig. 9a for $^{231}$Pa and 9b for $^{230}$Th). Similarly, we further decompose the contribution of bottom scavenging into those for $^{231}$Pa and $^{230}$Th (Figs. 9c and 9d, respectively). In Fig. 9a, we demonstrate that ocean transport solely from $^{231}$Pa (i.e., $^{231}$Pa(3D)/$^{230}$Th(1D)) can reproduce the overall distribution of the sedimentary $^{231}$Pa/$^{230}$Th ratios in CTRL_EXP (Fig. 6e). This result confirms that ocean transport of $^{231}$Pa primarily controls the distribution of sedimentary $^{231}$Pa/$^{230}$Th ratios, consistent with previous studies (Yu et al., 1996; Marchal et al., 2000). These previous studies suggest that the distribution of $^{231}$Pa mainly determines the global distribution of sedimentary $^{231}$Pa/$^{230}$Th ratios because the residence time of $^{231}$Pa is longer than that of $^{230}$Th.

Here, we further discuss how the ocean transport of $^{231}$Pa controls the distribution of sedimentary $^{231}$Pa/$^{230}$Th ratios. Since changes in sedimentary $^{231}$Pa correspond to particulate $^{231}$Pa changes in the bottom ocean, we focus the ocean transport effect on particulate $^{231}$Pa (Fig. 7c). Consistent with Fig. 9a, Fig. 7c indicates that ocean transport acts to decrease (increase) particulate $^{231}$Pa in lower (higher) latitudes. We also found that particulate $^{231}$Pa changes (Fig. 7c) are similar to those in dissolved $^{231}$Pa (Fig. 7a). Because most of $^{231}$Pa are in the dissolved phase, the advection of particulate $^{231}$Pa itself is very small compared with that of dissolved $^{231}$Pa, and ocean transport takes place mainly in the form of dissolved $^{231}$Pa. Therefore, it is interpreted that ocean transport first controls the dissolved $^{231}$Pa, and then the corresponding changes in particulate $^{231}$Pa take place so that the relationship between dissolved and particulate $^{231}$Pa (i.e., Eq. (5b)) is satisfied. In other words, the changes in particulate $^{231}$Pa take place as a result of changes in dissolved $^{231}$Pa. Therefore, we need to focus on the processes that control the dissolved $^{231}$Pa changes (Fig. 7a). As previously mentioned, in the case of no ocean transport (i.e., 1D_EXP), the dissolved $^{231}$Pa concentration near the seabed in lower latitudes becomes very high (Fig. S2a). Ocean transport, which includes both advection and diffusion, reduces high concentrations of dissolved $^{231}$Pa in low latitude oceans by transporting dissolved $^{231}$Pa from lower latitudes to higher latitudes. As a result of the change in the dissolved $^{231}$Pa (Fig. 7a), the changes in particulate $^{231}$Pa (Fig. 7c) also take place by satisfying Eq. (5b); this leads to lower sedimentary $^{231}$Pa/$^{230}$Th ratios in lower latitudes and higher ratios in higher latitudes (Figs. 7e and 7f).

Contrary to $^{231}$Pa, the influences of $^{230}$Th transport on sedimentary $^{231}$Pa/$^{230}$Th ratios have been usually regarded as small because $^{230}$Th is generally assumed to be scavenged very quickly everywhere. However, our results demonstrate that ocean transport of $^{230}$Th also affects the distribution of sediment $^{231}$Pa/$^{230}$Th to some extent. As a matter of fact, $^{230}$Th ocean transport acts in the opposite direction of $^{231}$Pa ocean transport, reducing the spatial contrast in sedimentary $^{231}$Pa/$^{230}$Th ratios (Fig. 9b). However, an exception is found in the Southern Ocean, where the $^{230}$Th ocean transport contributes to higher sedimentary $^{231}$Pa/$^{230}$Th ratios, in the same way as the $^{231}$Pa ocean transport. Because opal scavenges $^{231}$Pa more effectively than $^{230}$Th (Chase et al., 2002), $^{231}$Pa transported toward the Southern Ocean is expected to be quickly removed there due to

the high opal flux. Therefore, previous studies concluded that ocean transport of [231]Pa explains high sedimentary [231]Pa/[230]Th ratios in the Southern Ocean. On the other hand, in addition to ocean transport of [231]Pa, our results suggest that ocean transport of [230]Th also contributes to the high [231]Pa/[230]Th ratios in the Southern Ocean. This result implies that scavenging of [230]Th is not so efficient in the Southern Ocean as previously expected due to the dependence of scavenging efficiency on particle concentration. This interpretation is consistent with the high concentration of dissolved [230]Th in the Southern Ocean (Fig. 6b). Missiaen et al. (2020a) demonstrated that the dissolved [230]Th concentration in the Southern Ocean will increase if the effect of particle scavenging is halved and that most of this effect comes from POC and opal. This implies the scavenging of [230]Th is controlled also by the opal in the Southern Ocean. Together with their and our results, quantification about scavenging of [230]Th by opal in the Southern Ocean may be a key for a more accurate understanding of [231]Pa/[230]Th ratios in the global ocean.

Bottom scavenging promotes the removal of both [231]Pa and [230]Th near the seafloor and tends to cancel the influence of ocean transport. Namely, the bottom scavenging of [231]Pa reduces the contrast among sedimentary [231]Pa/[230]Th ratios (Fig. 9c), whereas the bottom scavenging of [230]Th increases this contrast (Fig. 9d). Because the influences of bottom scavenging of [231]Pa tends to be stronger than that of [230]Th, bottom scavenging overall results in reducing the contrast of [231]Pa/[230]Th ratios (Figs. 8e and 8f). Precisely speaking, the actual processes of the bottom scavenging effect on the sedimentary [231]Pa and [230]Th appear somewhat complicated compared with those of the ocean transport effect. The effect of the bottom scavenging is two-fold. First, extra particles in the bottom ocean lead to an increase of sedimentary [231]Pa and [230]Th (e.g., positive values near the bottom in low latitudes in Fig. 8c). Second, the bottom scavenging removes [231]Pa and [230]Th from the ocean, which reduces the concentration of dissolved [231]Pa and [230]Th in the ocean interior (Figs. 8a and 8b). The changes in dissolved-phase concentration then lead to changes in particulate-phase concentration in a way such that the Eq. (5b) is satisfied. The former leads to higher sedimentary [231]Pa and [230]Th, whereas the latter leads to lower sedimentary [231]Pa and [230]Th. Our results indicate that the former process becomes more important than the latter in the low latitudes, and the sedimentary [231]Pa increases there. In contrast, the latter dominates in the high latitudes, and the sedimentary [231]Pa decreases there by the bottom scavenging effect. The effect of bottom scavenging on [230]Th is also basically similar to [231]Pa.

## 4.4 Residence time of [231]Pa and [230]Th

Additional insights on the simulated distribution of [231]Pa/[230]Th ratios can be obtained from a comparison of CTRL_EXP with Siddall_EXP which reproduces sedimentary [231]Pa/[230]Th ratios (Fig. S4a) as realistically as does CTRL_EXP (Fig. 6e). In this subsection, we discuss this point by focusing on the difference in the residence time of [231]Pa and [230]Th between CTRL_EXP and Siddall_EXP. Assuming the mass balance of [231]Pa and [230]Th are in a steady state, we calculate the residence time of [231]Pa and [230]Th from the following formulas:

$$\tau^i = \frac{\int A_{total}^i dv}{F_{in}^i}, (11a)$$

$$F_{in}^i = \int \beta^i dv, (11b)$$

In Eq. (11a) and (11b), the integral domain is global and the parameters are described in Table 1. The residence times of $^{231}$Pa and $^{230}$Th are calculated to be 103 and 21 years, respectively, in CTRL_EXP, whereas they are 211 and 89 years, respectively, in Siddall_EXP (Table S2). By incorporating bottom scavenging and modifying the partition coefficient of $^{230}$Th, the modeled residence time in CTRL_EXP comes close to the previous estimate based on data: 111 years for $^{231}$Pa and 26 years for $^{230}$Th in Yu et al. (1996) and 130 years for $^{231}$Pa and 20 years for $^{230}$Th in Henderson and Anderson (2003). Because the reference partition coefficients for $^{231}$Pa of Siddall_EXP and that of CTRL_EXP are the same value (i.e., $K_{ref}^{Pa} = 1.0 \times 10^7$), the influence of ocean transport on $^{231}$Pa is identical in both experiments (Fig. 9a). Therefore, the difference in the $^{231}$Pa distribution between the model experiments must come from the bottom scavenging, which is included in CTRL_EXP but not in Siddall_EXP. The bottom scavenging reduces the residence time of $^{231}$Pa in CTRL_EXP (103 years) compared to Siddall_EXP (211 years). The difference in the $^{230}$Th distribution between CTRL_EXP and Siddall_EXP mainly comes from the difference in reference partition coefficients ($K_{ref}^{Th}$). The reference partition coefficient $K_{ref}^{Th}$ of CTRL_EXP, which depends on particle concentration, is larger than that of Siddall_EXP ($K_{ref}^{Th} = 6.0 \times 10^7$) in most of the ocean. Therefore, the contribution from the ocean transport of $^{230}$Th becomes larger in Siddall_EXP (Fig. S4b) than in CTRL_EXP (Fig. 6b). Together with additional contribution from the bottom scavenging effect on $^{230}$Th (Fig. 9d), the residence time of $^{230}$Th in CTRL_EXP (21 years) is shorter than that in Siddall_EXP (89 years). Since the residence time is overestimated for both $^{231}$Pa and $^{230}$Th in Siddall_EXP compared to CTRL_EXP, the distribution of sedimentary $^{231}$Pa/$^{230}$Th ratios in Siddall_EXP ends up similar to that in CTRL_EXP. The residence time in CTRL_EXP is similar to the residence time of their control simulation in Gu and Liu (2017), which does not include the bottom scavenging process. Therefore, total scavenging efficiency in the ocean is more important than the introduction of bottom scavenging to reproduce residence time. Compared with Siddall_EXP based on Siddall et al. (2005), our CTRL_EXP can realistically simulate not only oceanic distribution of $^{231}$Pa and $^{230}$Th but also their residence time by introducing the bottom scavenging and the dependence of scavenging efficiency on particulate concentration.

## 4.5 Remaining issues

Although our model was able to generally reproduce the basin-scale distributions of $^{231}$Pa and $^{230}$Th, there are still some mismatches between the model results and observations. For dissolved $^{231}$Pa, introducing bottom scavenging helped to reproduce the concentrations seen in the data at depths below 3000m (Fig. 2). However, the model tends to simulate lower concentration than the observations below 3000m in Figs. 2c and 2d, which needs to be improved. The improvement was not possible simply by reducing the bottom scavenging (i.e., specifying smaller $K_{bottom}^{Pa}$ than in Figs. 2c and 2d), therefore more fundamental improvement appears to be required. The dissolved $^{230}$Th simulated in CTRL_EXP (Fig. 5b) also tends to underestimate the observed concentration near the sea bottom. One possibility is that our treatment of the nepheloid layer (i.e., the thickness of the ocean deepest layer) may be too simple and needs to be modified so that the thickness of the nepheloid layer is more realistically specified. The introduction of more realistic bottom scavenging and the consideration of the effects

of particles from the continental shelf and hydrothermal vents may also help to improve the model-data agreement for both $^{231}$Pa and $^{230}$Th.

In this study, particle fields were not calculated in the model but specified as boundary conditions in our approach where the specified distribution of biological particles is taken from satellite-based estimation. Since the bias of the particle field affects the distribution of $^{231}$Pa and $^{230}$Th, our approach has advantages over the other studies where the particles are explicitly simulated in the model. However, satellite-based estimation referenced here may also contain some errors, and understanding about the influence of the particle field on sedimentary $^{231}$Pa/$^{230}$Th ratios, which was not seriously discussed in this study, is also important as the previous studies pointed out (e.g., Missiaen et al., 2020b; Dutay et al., 2009; van Hulten et al., 2018).

To reconstruct past sedimentation flux, $^{230}$Th normalization was used. Recently, the influence of lithogenic and authigenic $^{230}$Th on $^{230}$Th in sediments was evaluated (Missiaen et al., 2018; Costa et al., 2020). This is not a direct topic of our study, but we need to care about such processes which also affect the ratio of $^{231}$Pa and $^{230}$Th obtained from marine sediments. As more observational data and their modeling become available, we expect to make further progress in quantitative understanding of the processes governing $^{231}$Pa and $^{230}$Th in the ocean.

## 5 Summary and concluding remarks

In this study, we performed OGCM experiments that incorporated the bottom scavenging and the dependence of scavenging efficiency on particle concentration together with the water-column reversible scavenging. We quantitatively evaluated the processes that determine the global distribution of sedimentary $^{231}$Pa/$^{230}$Th ratios, which is used as a proxy for the strength of paleo-ocean circulation.

First, we performed an OGCM experiment using the same model settings and parameters as Siddall et al. (2005), which only introduced the water-column reversible scavenging (Siddall_EXP). In Siddall_EXP, the simulated concentrations of $^{231}$Pa and $^{230}$Th increase with depth, consistent with data; however, this experiment significantly overestimated the concentrations observed in the deep ocean. By incorporating bottom scavenging in nepheloid layers following Rempfer et al. (2017) (BTM_EXP), we reduced this overestimation and successfully reproduced the vertical profile of dissolved $^{231}$Pa. However, this experiment had difficulty in reproducing the observed vertical profile of dissolved $^{230}$Th. Therefore, we modified the parameters associated with the strength of water-column scavenging (i.e., $K_{ref}$: the reference partition coefficient for sinking particles) with the consideration of the bottom scavenging (KREF_EXP). When we increased the reference partition coefficient of $^{230}$Th ($K_{ref}^{Th} = 6.0 \times 10^{7}$) from that used in the Siddall_EXP with the consideration of bottom scavenging ($K_{bottom}^{Th} = 1.0 \times 10^{7}$), dissolved $^{230}$Th was found to be more realistically simulated, but significant underestimation in the Southern Ocean remained. We found that the underestimation in the Southern Ocean can be improved by introducing dependence of $K_{ref}$ on particle concentration which was used in Henderson et al. (1999) (PCE_EXP). Although most of the

previous [231]Pa and [230]Th model results showed significant overestimation in the deep ocean (e.g., Siddall et al., 2005; Dutay et al., 2009; Gu and Liu, 2017; van Hulten et al., 2018), our best OGCM simulation considering the reversible scavenging, bottom scavenging, and the dependence of scavenging efficiency on particle concentration (CTRL_EXP) can reproduce the distributions of dissolved [231]Pa and [230]Th consistently with GEOTRACES data, together with the realistic distribution of sedimentary [231]Pa/[230]Th ratios.

We also made a quantitative assessment about the processes that determine the global distribution of sedimentary [231]Pa/[230]Th ratios by decomposing the processes affecting the sediment [231]Pa/[230]Th ratios into three parts: water-column scavenging, ocean transport (advection and diffusion), and bottom scavenging. We confirmed that the global sedimentary [231]Pa/[230]Th ratios in our best model (CTRL_EXP) are primarily determined by ocean transport of [231]Pa, as shown in previous studies. Contrary to [231]Pa, ocean transport of [230]Th tends to reduce the spatial contrast of sedimentary [231]Pa/[230]Th ratios. However, we found that this is not the case for the Southern Ocean; [230]Th advection increases the sedimentary [231]Pa/[230]Th ratios in the Southern Ocean and strengthens the observed high [231]Pa/[230]Th ratios there. This means that not only [231]Pa advection but also [230]Th advection contributes to the high [231]Pa/[230]Th ratios in the Southern Ocean. This result implies that scavenging of [230]Th is not much efficient in the Southern Ocean as conventionally thought when we consider the dependence of scavenging efficiency on particle concentration. We also show that bottom scavenging works opposite to ocean transport and decreases the spatial contrast of [231]Pa/[230]Th ratios; bottom scavenging promotes the removal of [231]Pa near the sea bottom more efficiently than that of [230]Th, and the total effect of bottom scavenging reduces spatial contrasts of the [231]Pa/[230]Th ratios. Our best simulation shows the realistic residence times of [231]Pa and [230]Th, but simulation without bottom scavenging and dependence of scavenging efficiency on particle concentration significantly overestimates the residence times for both [231]Pa and [230]Th in spite of similar distribution of sedimentary [231]Pa/[230]Th ratios to our best simulation.

The model developed in this study is useful not only for simulating [231]Pa/[230]Th ratios in the present-day ocean but also in different climates such as glacial periods. Our OGCM experiments using the present-day physical fields can clarify the processes governing the global distribution of sedimentary [231]Pa/[230]Th ratios. A similar analysis using the physical ocean fields during glacial periods may help climate scientists to understand the mechanisms for glacial changes in the sedimentary [231]Pa/[230]Th ratio observed in sediment cores. Although simulated sedimentary [231]Pa/[230]Th under glacial times are also discussed in a 2D model (Lippold et al., 2012) and recently in a 3D model (Gu et al., 2020), there is insufficient discussion of the mechanism of change in the three-dimensional distribution. Simulation of [231]Pa/[230]Th ratios under glacial climates (e.g., Oka et al., 2011; Kobayashi and Oka, 2018) is an exciting avenue of future study.

**Code and data availability**

The [231]Pa/[230]Th model code and data used to produce the results in this study are available at the repository website Zenodo: https://doi.org/10.5281/zenodo.4600287 (Sasaki et al., 2021a) and https://doi.org/10.5281/zenodo.4655882 (Sasaki et al., 2021b), respectively. COCO is an ocean component of MIROC and the code of COCO4 is included as a part of MIROC-

ES2L. The source code of MIROC-ES2L can be obtained from https://doi.org/10.5281/zenodo.3893386 (Ohgaito et al., 2020).

## Supplement

The supplement related to this article is available online at: https://doi.org/xxxx.

## Author contributions

All the authors contributed to the interpretation of the simulation results. Y.S. performed the numerical simulations. A.O. designed and supervised the study. Y.S. and H.K. analyzed the results. Y.S. wrote the first draft and the final draft was prepared with the inputs from all the coauthors.

## Competing interests

The authors declare that they have no conflict of interest.

## Acknowledgments

The authors acknowledge many constructive comments from reviewers, which significantly improves the manuscript. This work was supported by JSPS KAKENHI Grant Number JP19H01963.

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

**Figure 1**. (a) Dissolved $^{231}$Pa along 30°W in the Atlantic Ocean and (b) its vertical profile (the latitudinal mean along 30°W in the Atlantic Ocean) in Siddall_EXP. (c, d) Same as Figs. 1a and 1b except for $^{230}$Th. The colored circles in Figs. 1a and 1c represent data from the Atlantic GEOTRACES data (GA02 and GIPY05; Schlitzer et al., 2018). The green and orange circles in Figs. 1b and 1d represent the GA02 data and simulation results.

**Figure 2**. (a, c, e) Dissolved $^{231}$Pa along 30°W in the Atlantic Ocean and (b, d, f) its vertical profile (the latitudinal mean along 30°W in the Atlantic Ocean) in BTM EXP. $K_{bottom}^{Pa}$ is set to $5.0 \times 10^4$ in Figs. 2a and 2b, $5.0 \times 10^5$ in Figs. 2c and 2d, and $5.0 \times 10^6$ in Figs. 2e and 2f. The colored circles in Figs. 2a, 2c, and 2e represent data from the Atlantic GEOTRACES data (GA02 and GIPY05; Schlitzer et al., 2018). The green and orange circles in Figs. 2b, 2d, and 2f represent the GA02 data and simulation results.

**Figure 3**. (a, c, e, g) Dissolved $^{230}$Th along 30°W in the Atlantic Ocean and (b, d, f, h) its vertical profile (the latitudinal mean along 30°W in the Atlantic Ocean) in BTM_EXP are plotted. $K_{bottom}^{Th}$ is set to $5.0 \times 10^5$ in Figs. 3a and 3b, $1.0 \times 10^6$ in Figs. 3c and 3d, $5.0 \times 10^6$ in Figs. 3e and 3f, and $1.0 \times 10^7$ in Figs. 3g and 3h. The colored circles in Figs. 3a, 3c, 3e, and 3g represent data from the Atlantic GEOTRACES data (GA02 and GIPY05; Schlitzer et al., 2018). The green and orange circles in Figs. 3b, 3d, 3f, and 3h represent the GA02 data and simulation results.

**Figure 4**. The vertical profile of dissolved $^{230}$Th (the latitudinal mean along 30°W in the Atlantic Ocean) in various simulations of KREF_EXP: (a) R2_B20, (b) R2_B10, (c) R2_B5, (d) R4_B10, (e) R4_B5, and (f) R6_B5. The green and orange circles in Figs. 4a–4f represent the Atlantic GEOTRACES data (GA02; Schlitzer et al., 2018) and simulation results. Figure 4g summarizes the choice of parameters (i.e., $K_{ref}^{Th}$ and $K_{bottom}^{Th}$) in each simulation.

**Figure 5**. Dissolved $^{230}$Th along 30°W in the Atlantic Ocean in (a) R6_B5 of the KREF_EXP and (b) PCE_EXP. (c) Reference partition coefficient (K_ref) along 30°W in the Atlantic Ocean in PCE_EXP. (d) The vertical profile of dissolved $^{230}$Th (the latitudinal mean along 30°W in the Atlantic Ocean) in R6_B5 of KREF_EXP and PCE_EXP. The colored circles in Figs. 5a and 5b represent data from the Atlantic GEOTRACES data (GA02 and GIPY05; Schlitzer et al., 2018). The green, yellow, and orange circles in Fig. 5d represent the GA02 data and KREF_EXP and PCE_EXP simulation results.

**Figure 6.** (a) Dissolved $^{231}$Pa, (b) dissolved $^{230}$Th, (c) particulate $^{231}$Pa, and (d) particulate $^{230}$Th along 30°W in the Atlantic Ocean in CTRL_EXP. (e) Sedimentary $^{231}$Pa/$^{230}$Th ratios normalized by the production ratio of 0.093 in CTRL_EXP. The colored circles represent observational data. Dissolved $^{231}$Pa and $^{230}$Th data are taken from the Atlantic GEOTRACES data (GA02 and GIPY05; Schlitzer et al., 2018). Particulate $^{231}$Pa and $^{230}$Th data are taken from the following references (Colley et al., 1995; Moran et al., 1997; Moran et al., 2001; Rutgers van der Loeff and Berger, 1993; Vogler et al., 1998; Walter et al., 1997; Cochran et al., 1987; Moran et al., 2002; Guo et al., 1995). The data of sedimentary $^{231}$Pa/$^{230}$Th ratios are taken from the following references (Mangini & Sonntag, 1977; Muller & Mangini, 1980; Anderson et al., 1983; Shimmield et al., 1986; Schmitz et al., 1986; Yang et al., 1986; Shimmield & Price, 1988; Yong Lao et al., 1992; François et al., 1993; Frank et al., 1994; Frank, 1996; Bradtmiller et al., 2014, Luo et al., 2010, and their supplemental data).

**Figure 7.** The difference between 3D_EXP and 1D_EXP (i.e., 3D_EXP minus 1D_EXP, which represents for ocean transport effect) of (a) dissolved $^{231}$Pa, (b) dissolved $^{230}$Th, (c) particulate $^{231}$Pa, and (d) particulate $^{230}$Th along 30°W in the Atlantic Ocean. (e) The difference between 3D_EXP and 1D_EXP of sedimentary $^{231}$Pa/$^{230}$Th ratios is normalized by the production ratio of 0.093.

**Figure 8.** The difference between CTRL_EXP and 3D_EXP (i.e., CTRL_EXP minus 3D_EXP, which represents for bottom scavenging effect) of (a) dissolved $^{231}$Pa, (b) dissolved $^{230}$Th, (c) particulate $^{231}$Pa, and (d) particulate $^{230}$Th along 30°W in the Atlantic Ocean. (e) The difference between CTRL_EXP and 3D_EXP of the sedimentary $^{231}$Pa/$^{230}$Th ratios is normalized by the production ratio of 0.093.

**Figure 9.** Sedimentary $^{231}$Pa/$^{230}$Th ratios normalized by the production ratio of 0.093 in CTRL_EXP decomposed into contributions from (a) ocean transport solely from $^{231}$Pa (i.e., $^{231}$Pa(3D)/$^{230}$Th(1D)), (b) ocean transport solely from $^{230}$Th (i.e., $^{231}$Pa(1D)/$^{230}$Th(3D)), (c) bottom scavenging solely from $^{231}$Pa (i.e., $^{231}$Pa(CTRL)/$^{230}$Th(3D) minus $^{231}$Pa(3D)/$^{230}$Th(3D)), and (d) bottom scavenging solely from $^{230}$Th (i.e., $^{231}$Pa(3D)/$^{230}$Th(CTRL) minus $^{231}$Pa(3D)/$^{230}$Th(3D)).

**Table 1.** Parameters of the $^{231}$Pa and $^{230}$Th model.

**Table 2.** Equilibrium partition coefficients in experiments Siddall_EXP and CTRL_EXP.

**Table 3.** Processes considered in additional experiments. A circle means that the process is considered, and a cross means that it is not considered.

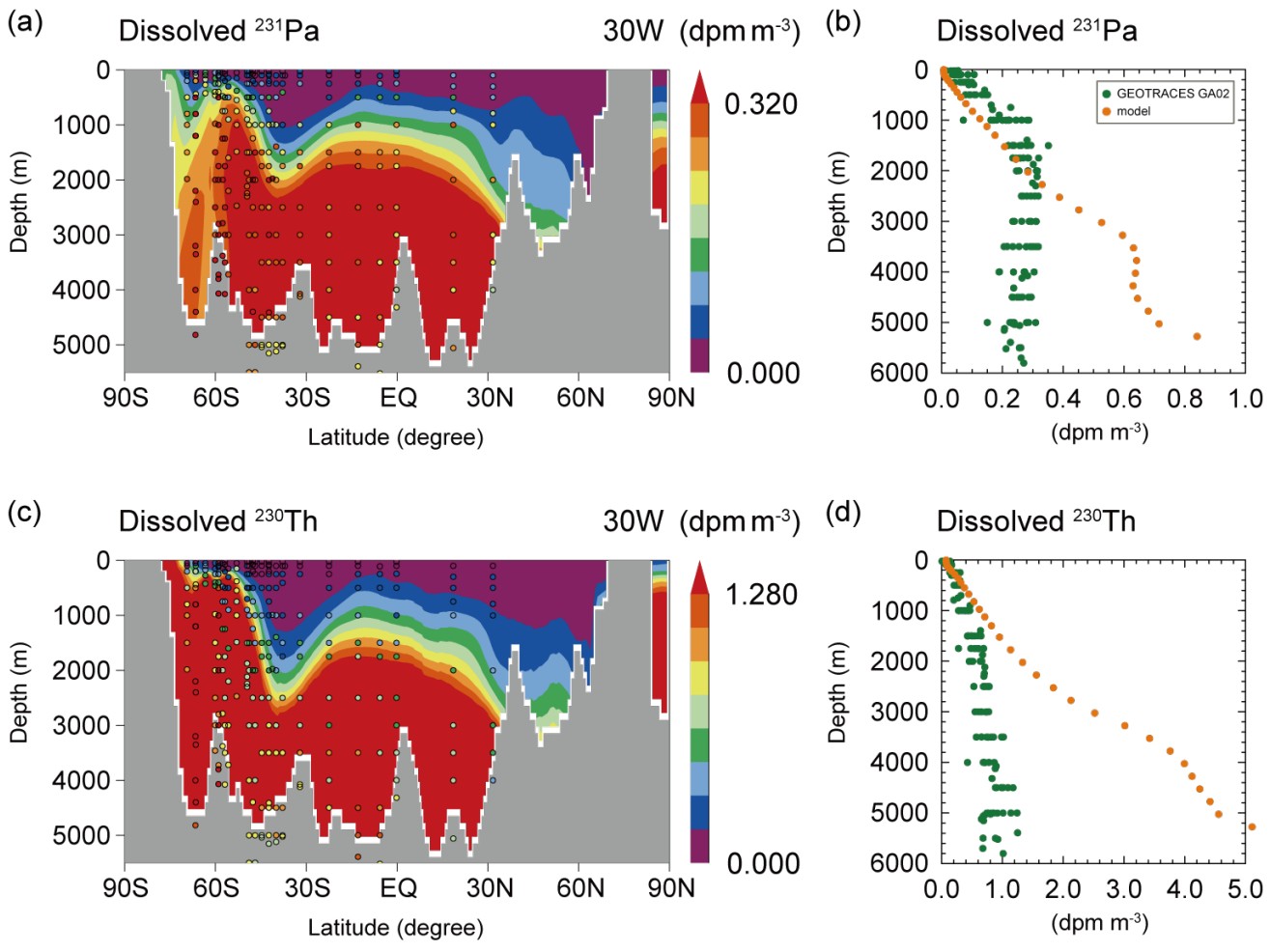

Figure 1. (a) Dissolved $^{231}$Pa along 30°W in the Atlantic Ocean and (b) its vertical profile (the latitudinal mean along 30°W in the Atlantic Ocean) in Siddall_EXP. (c, d) Same as Figs. 1a and 1b except for $^{230}$Th. The colored circles in Figs. 1a and 1c represent data from the Atlantic GEOTRACES data (GA02 and GIPY05; Schlitzer et al., 2018). The green and orange circles in Figs. 1b and 1d represent the GA02 data and simulation results, respectively.

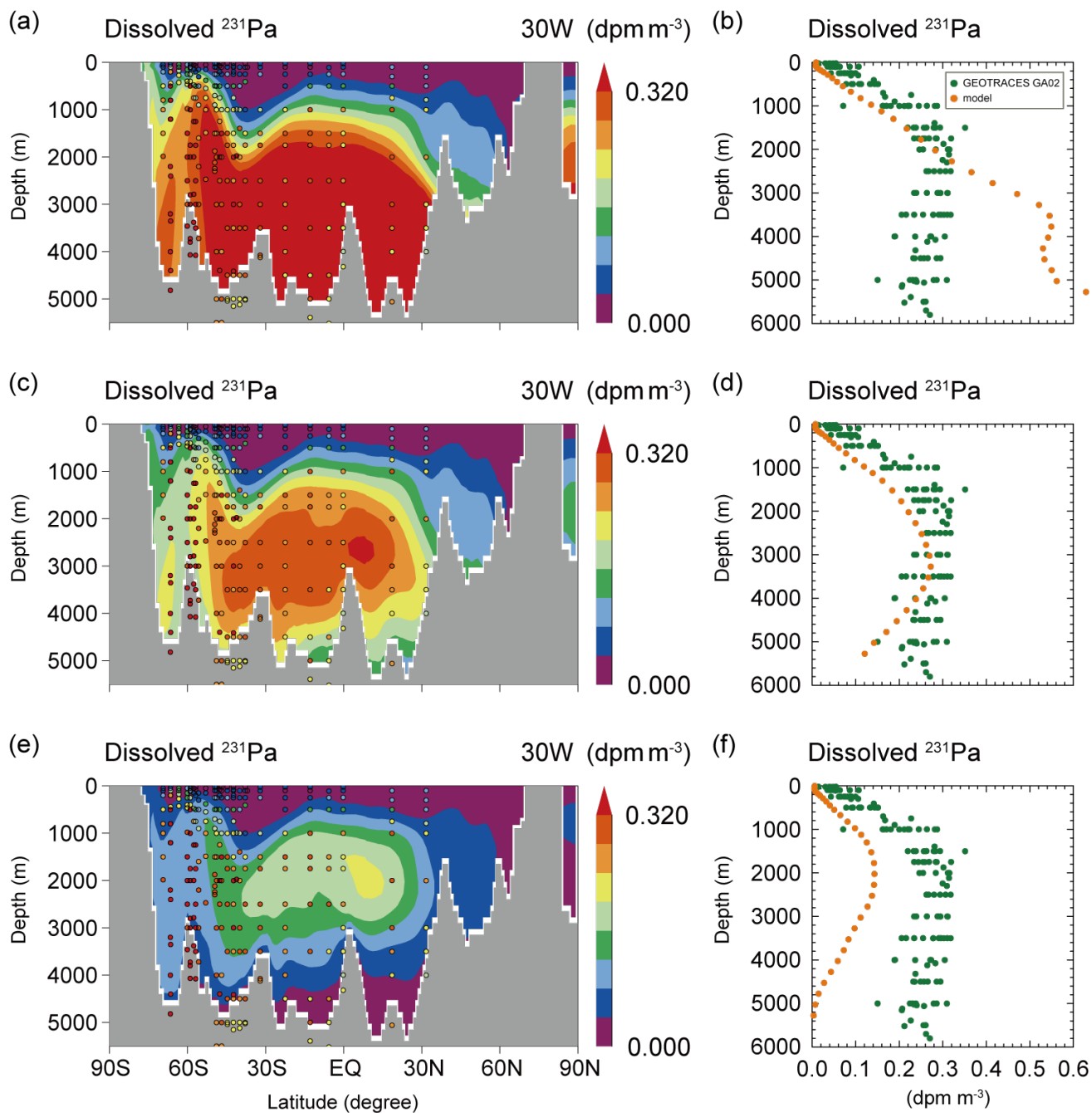

Figure 2. (a, c, e) Dissolved $^{231}$Pa along 30°W in the Atlantic Ocean and (b, d, f) its vertical profile (the latitudinal mean along 30°W in the Atlantic Ocean) in BTM_EXP. $K_{bottom}^{Pa}$ is set to $5.0 \times 10^4$ in Figs. 2a and 2b, $5.0 \times 10^5$ in Figs. 2c and 2d, and $5.0 \times 10^6$ in Figs. 2e and 2f. The colored circles in Figs. 2a, 2c, and 2e represent data from the Atlantic GEOTRACES data (GA02 and GIPY05; Schlitzer et al., 2018). The green and orange circles in Figs. 2b, 2d, and 2f represent the GA02 data and simulation results.

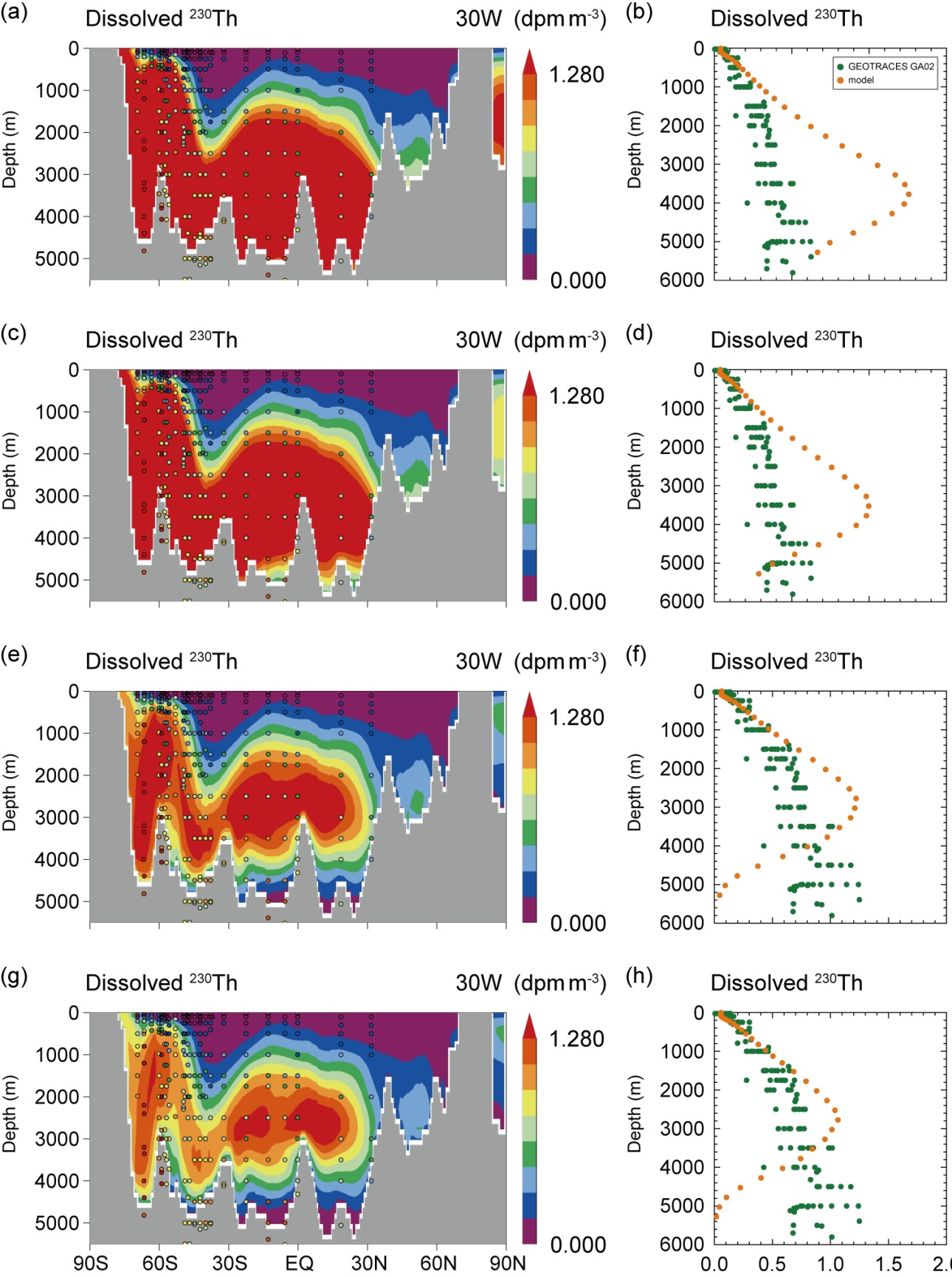

Figure 3. (a, c, e, g) Dissolved $^{230}$Th along 30°W in the Atlantic Ocean and (b, d, f, h) its vertical profile (the latitudinal mean along 30°W in the Atlantic Ocean) in BTM_EXP are plotted. $K_{bottom}^{Th}$ is set to 5.0×10$^5$ in Figs. 3a and 3b, 1.0×10$^6$ in Figs. 3c and 3d, 5.0×10$^6$ in Figs. 3e and 3f, and 1.0×10$^7$ in Figs. 3g and 3h. The colored circles in Figs. 3a, 3c, 3e, and 3g represent data from the Atlantic GEOTRACES data (GA02 and GIPY05; Schlitzer et al., 2018). The green and orange circles in Figs. 3b, 3d, 3f, and 3h represent the GA02 data and simulation results.

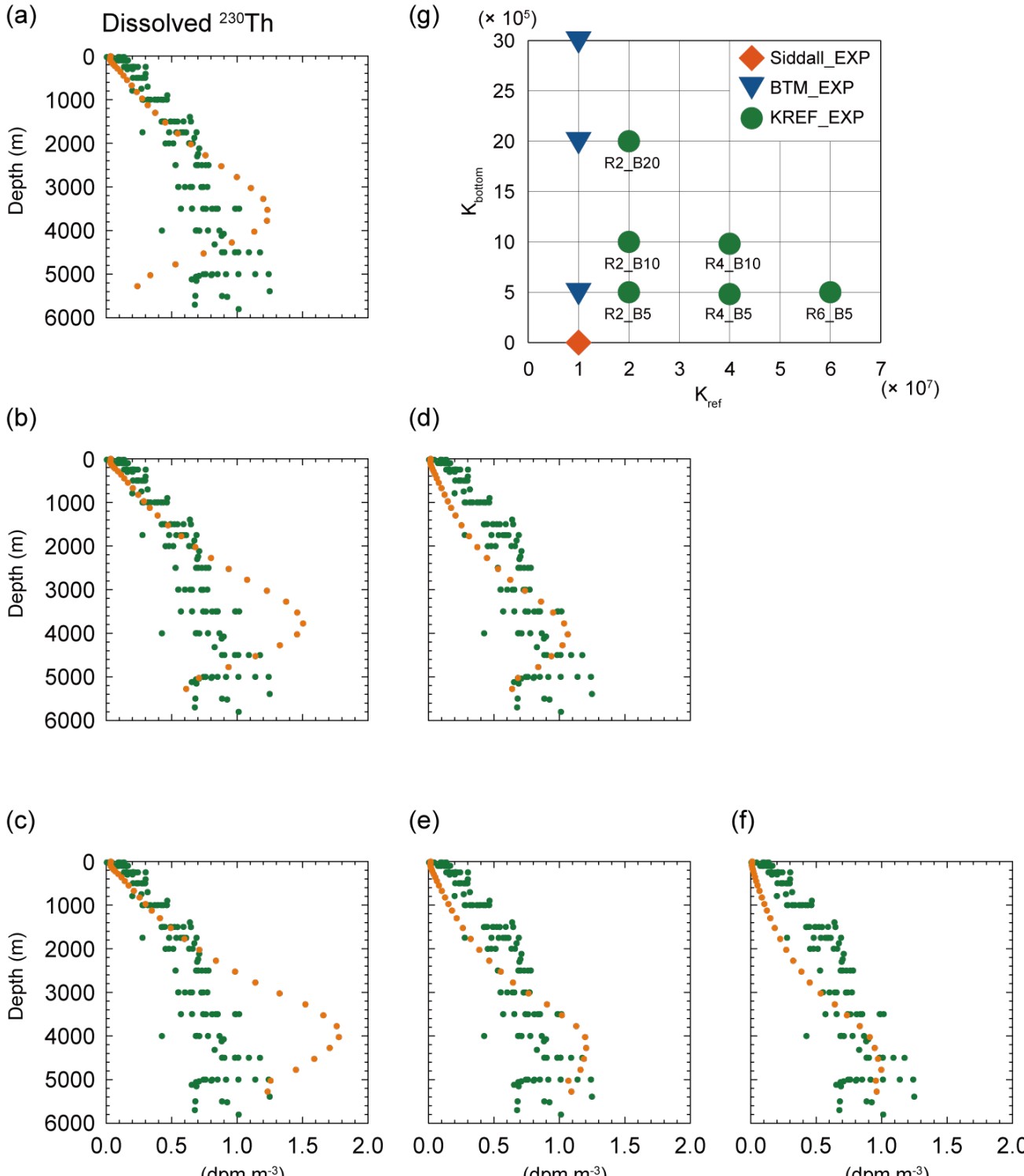

Figure 4. The vertical profile of dissolved $^{230}$Th (the latitudinal mean along 30°W in the Atlantic Ocean) in various simulations of KREF_EXP: (a) R2_B20, (b) R2_B10, (c) R2_B5, (d) R4_B10, (e) R4_B5, and (f) R6_B5. The green and orange circles in Figs. 4a–4f represent the Atlantic GEOTRACES data (GA02; Schlitzer et al., 2018) and simulation results. Figure 4g summarizes the choice of parameters (i.e., $K_{ref}^{Th}$ and $K_{bottom}^{Th}$) in each simulation.

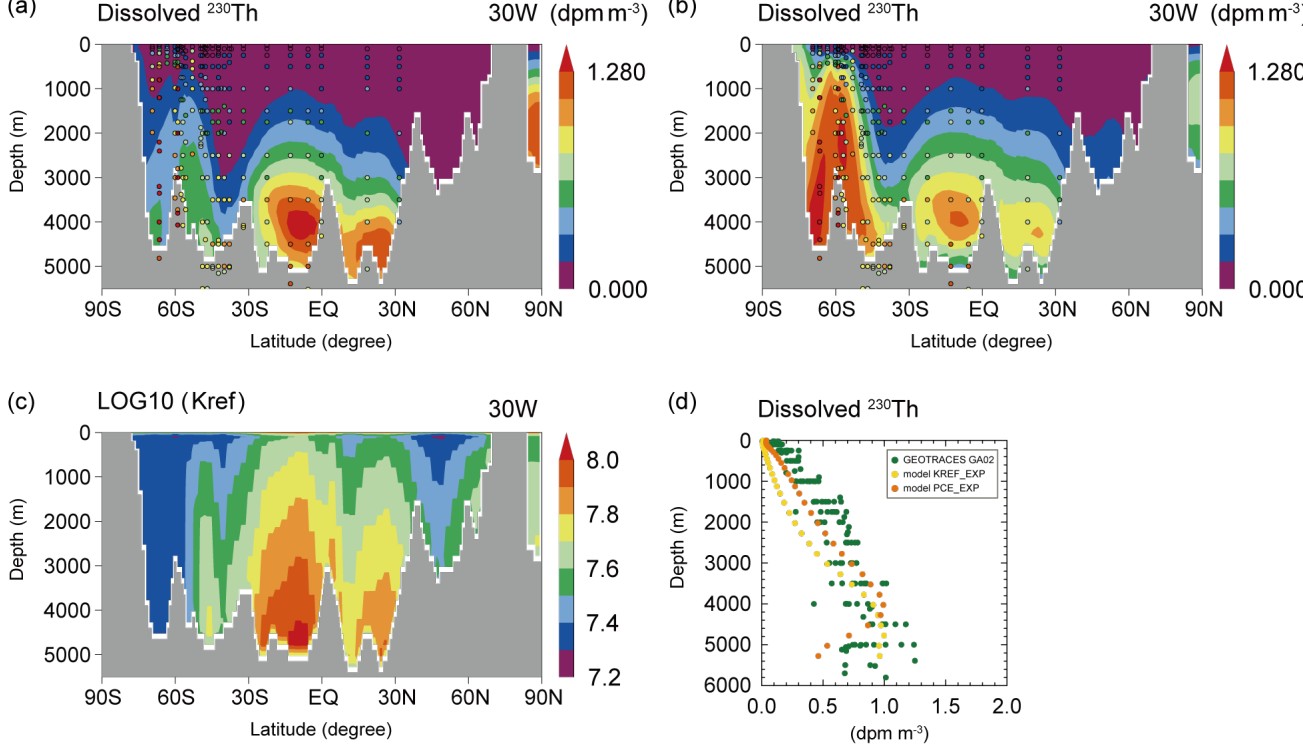

Figure 5. Dissolved $^{230}$Th along 30°W in the Atlantic Ocean in (a) R6_B5 of the KREF_EXP and (b) PCE_EXP. (c) Reference partition coefficient ($K_{ref}$) along 30°W in the Atlantic Ocean in PCE_EXP. (d) The vertical profile of dissolved $^{230}$Th (the latitudinal mean along 30°W in the Atlantic Ocean) in R6_B5 of KREF_EXP and PCE_EXP. The colored circles in Figs. 5a and 5b represent data from the Atlantic GEOTRACES data (GA02 and GIPY05; Schlitzer et al., 2018). The green, yellow, and orange circles in Fig. 5d represent the GA02 data and KREF_EXP and PCE_EXP simulation results.

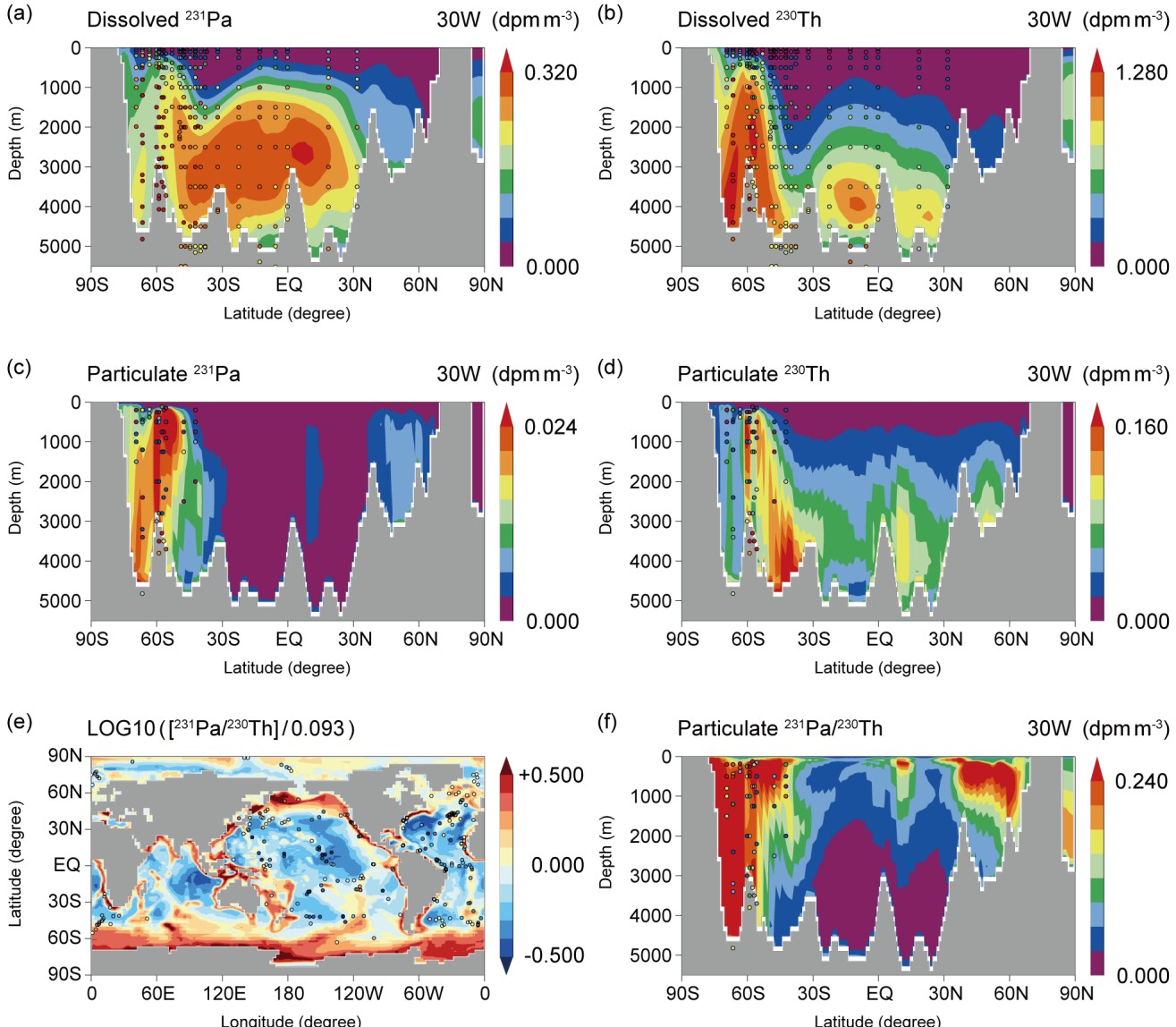

Figure 6. (a) Dissolved [231]Pa, (b) dissolved [230]Th, (c) particulate [231]Pa, and (d) particulate [230]Th along 30°W in the Atlantic Ocean in CTRL_EXP. (e) Sedimentary [231]Pa/[230]Th ratios normalized by the production ratio of 0.093 in CTRL_EXP. The colored circles represent observational data. Dissolved [231]Pa and [230]Th data are taken from the Atlantic GEOTRACES data (GA02 and GIPY05; Schlitzer et al., 2018). Particulate [231]Pa and [230]Th data are taken from the following references (Colley et al., 1995; Moran et al., 1997; Moran et al., 2001; Rutgers van der Loeff and Berger, 1993; Vogler et al., 1998; Walter et al., 1997; Cochran et al., 1987; Moran et al., 2002; Guo et al., 1995). The data of sedimentary [231]Pa/[230]Th ratios are taken from the following references (Mangini & Sonntag, 1977; Muller & Mangini, 1980; Anderson et al., 1983; Shimmield et al., 1986; Schmitz et al., 1986; Yang et al., 1986; Shimmield & Price, 1988; Yong Lao et al., 1992; François et al., 1993; Frank et al., 1994; Frank, 1996; Bradtmiller et al., 2014, Luo et al., 2010, and their supplemental data).

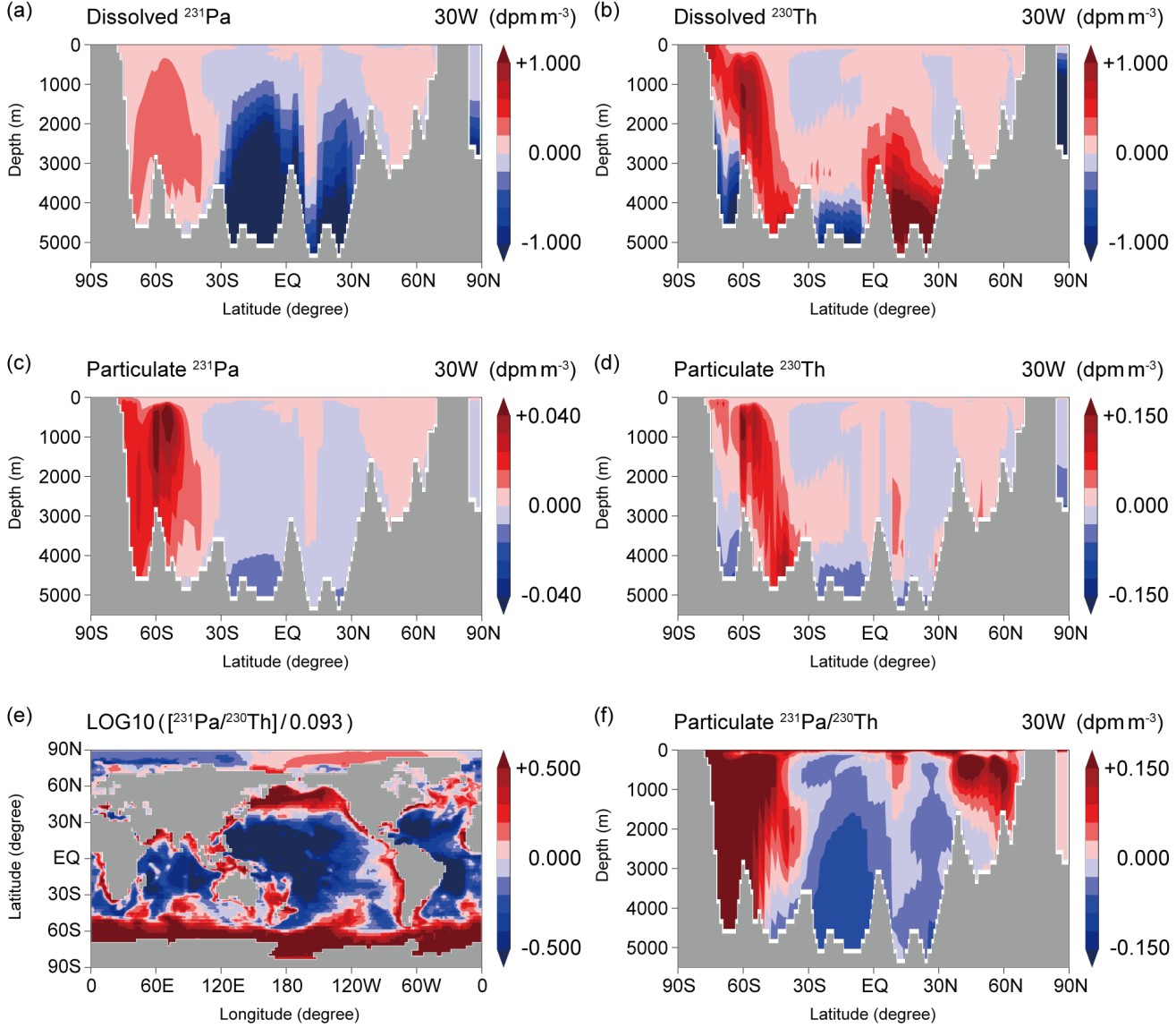

Figure 7. The difference between 3D_EXP and 1D_EXP (i.e., 3D_EXP minus 1D_EXP, which represents for ocean transport effect) of (a) dissolved $^{231}$Pa, (b) dissolved $^{230}$Th, (c) particulate $^{231}$Pa, and (d) particulate $^{230}$Th along 30°W in the Atlantic Ocean. (e) The difference between 3D_EXP and 1D_EXP of sedimentary $^{231}$Pa/$^{230}$Th ratios normalized by the production ratio of 0.093.

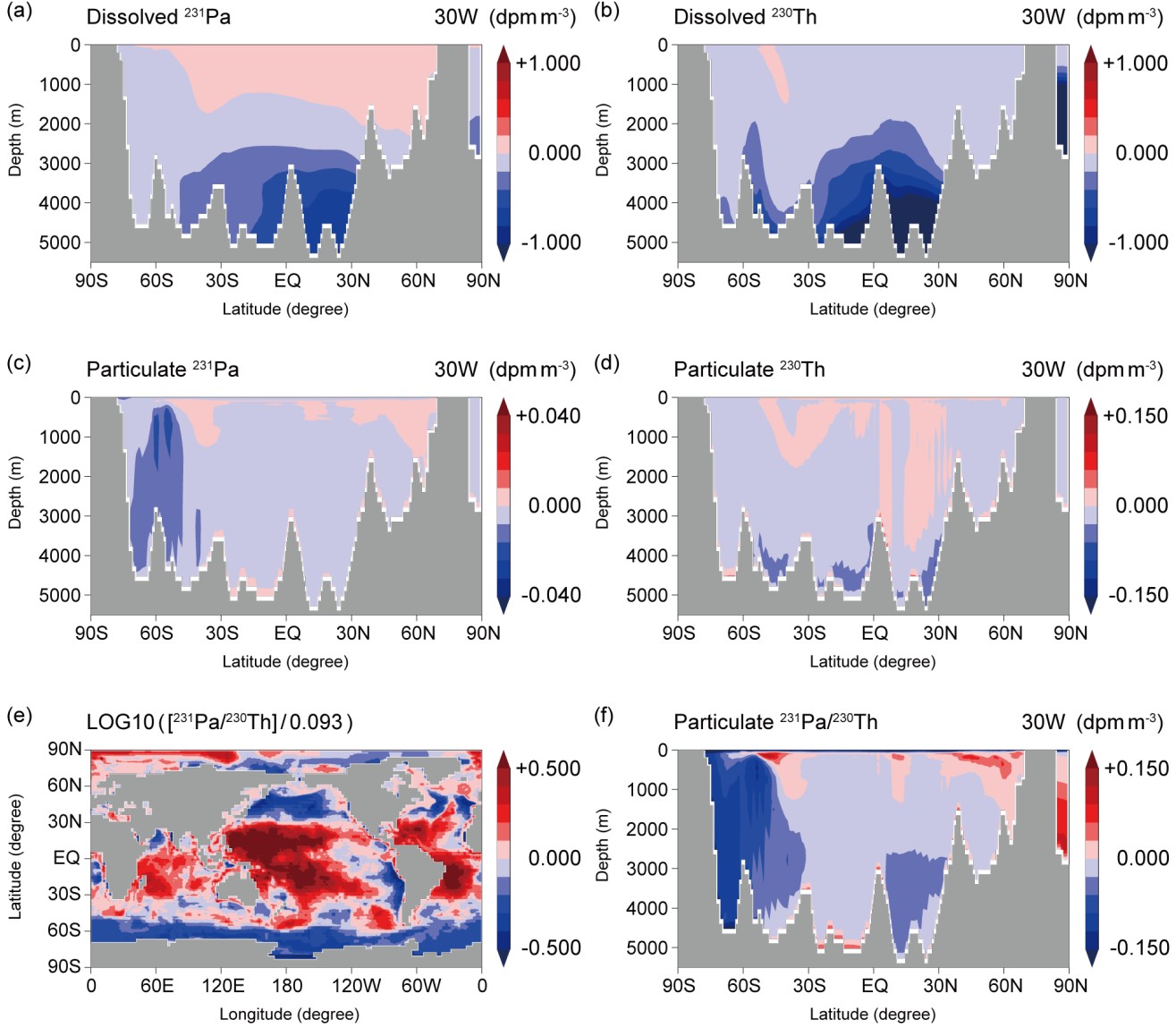

Figure 8. The difference between CTRL_EXP and 3D_EXP (i.e., CTRL_EXP minus 3D_EXP, which represents for bottom scavenging effect) of (a) dissolved $^{231}$Pa, (b) dissolved $^{230}$Th, (c) particulate $^{231}$Pa, and (d) particulate $^{230}$Th along 30°W in the Atlantic Ocean. (e) The difference between CTRL_EXP and 3D_EXP of the sedimentary $^{231}$Pa/$^{230}$Th ratios normalized by the production ratio of 0.093.

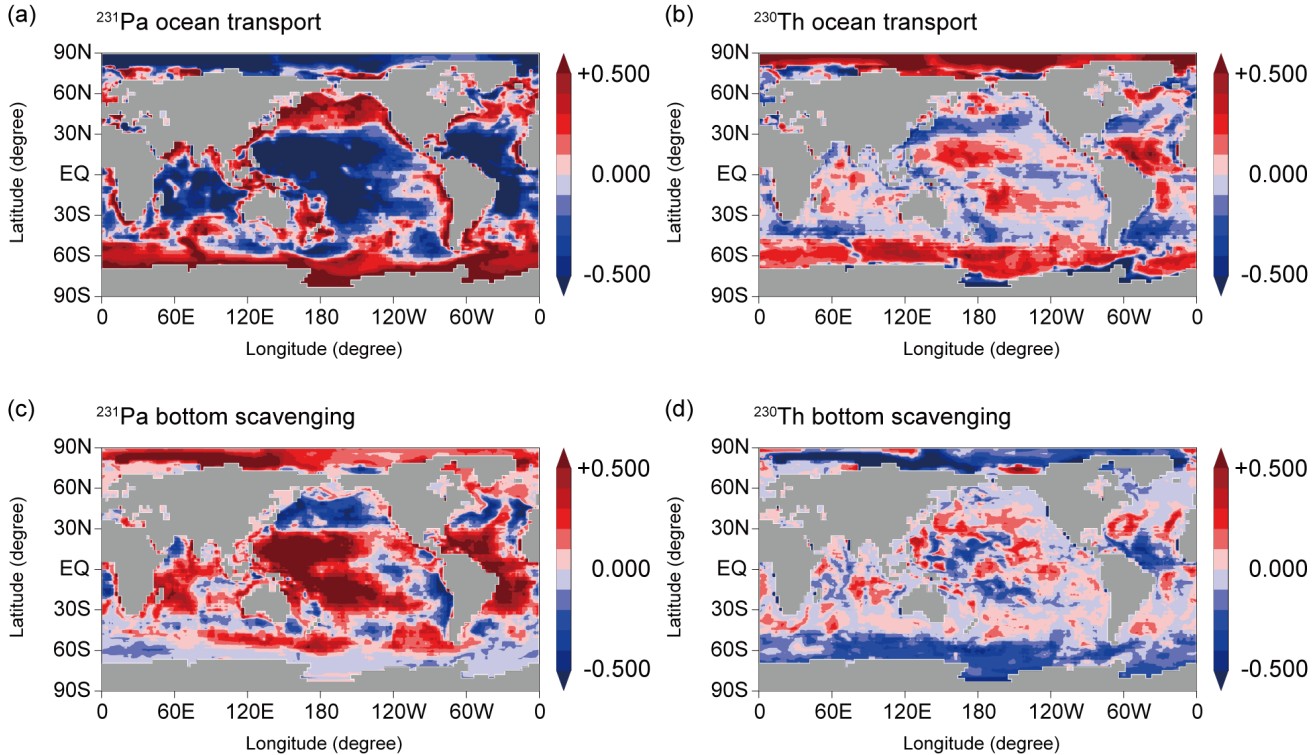

Figure 9. Sedimentary $^{231}$Pa/$^{230}$Th ratios normalized by the production ratio of 0.093 in CTRL_EXP decomposed into contributions from (a) ocean transport solely from $^{231}$Pa (i.e., $^{231}$Pa(3D)/$^{230}$Th(1D)), (b) ocean transport solely from $^{230}$Th (i.e., $^{231}$Pa(1D)/$^{230}$Th(3D)), (c) bottom scavenging solely from $^{231}$Pa (i.e., $^{231}$Pa(CTRL)/$^{230}$Th(3D) minus $^{231}$Pa(3D)/$^{230}$Th(3D)), and (d) bottom scavenging solely from $^{230}$Th (i.e., $^{231}$Pa(3D)/$^{230}$Th(CTRL) minus $^{231}$Pa(3D)/$^{230}$Th(3D)).

| Variable | Symbol | Value | Units |
|---|---|---|---|
| $^{231}$Pa production from $^{235}$U decay | $\beta^{Pa}$ | $2.33 \times 10^{-3}$ | $dpm\ m^{-3}\ yr^{-1}$ |
| $^{230}$Th production from $^{234}$U decay | $\beta^{Th}$ | $2.52 \times 10^{-2}$ | $dpm\ m^{-3}\ yr^{-1}$ |
| Decay constant of $^{231}$Pa | $\lambda^{Pa}$ | $2.13 \times 10^{-5}$ | $yr^{-1}$ |
| Decay constant of $^{230}$Th | $\lambda^{Th}$ | $9.12 \times 10^{-6}$ | $yr^{-1}$ |
| Sinking velocity of particles | $w_s$ | 1000 | $m\ yr^{-1}$ |
| Thickness of euphotic zone | $z_0$ | 100 | m |
| Penetration depth of CaCO$_3$ | $z_p$ | 2000 | m |
| Dissolution constant of opal | $B$ | 0.12 | $°C^{-1}\ yr^{-1}$ |
| Minimum temperature of sea water | $T_0$ | $-2$ | °C |
| Dissolution rate of POC | $\varepsilon$ | 0.858 | - |
| Total activity of $^{231}$Pa or $^{230}$Th | $A_{total}$ | variable | $dpm\ m^{-3}$ |
| Activity of dissolved $^{231}$Pa or $^{230}$Th | $A_d$ | variable | $dpm\ m^{-3}$ |
| Activity of particle $^{231}$Pa or $^{230}$Th | $A_p$ | variable | $dpm\ m^{-3}$ |
| Ratio of particle concentration to fluid density | $C$ | variable | - |

Table 1. Parameters of the $^{231}$Pa and $^{230}$Th model.

| Experiment | Siddall_EXP | | CTRL_EXP | |
| --- | --- | --- | --- | --- |
| | $^{231}$Pa | $^{230}$Th | $^{231}$Pa | $^{230}$Th |
| $K_{ref}$ | $1.0 \times 10^7$ | $1.0 \times 10^7$ | $1.0 \times 10^7$ | $\left(\dfrac{C_{total}}{10^{-7}}\right)^{-0.42} \times 10^7$ |
| $K_{CaCO_3}$ | $K_{ref}\,/\,40$ | $K_{ref}$ | $K_{ref}\,/\,40$ | $K_{ref}$ |
| $K_{opal}$ | $K_{ref}\,/\,6$ | $K_{ref}\,/\,20$ | $K_{ref}\,/\,6$ | $K_{ref}\,/\,20$ |
| $K_{POC}$ | $K_{ref}$ | $K_{ref}$ | $K_{ref}$ | $K_{ref}$ |
| $K_{bottom}$ | 0 | 0 | $5.0 \times 10^5$ | $5.0 \times 10^5$ |

Table 2. Equilibrium partition coefficients in experiments Siddall_EXP and CTRL_EXP.

| Experiment | Water-column reversible scavenging | Bottom scavenging | Ocean transport |
|---|---|---|---|
| CTRL_EXP | O | O | O |
| 3D_EXP | O | X | O |
| 1D_EXP | O | X | X |

Table 3. Processes considered in additional experiments. A circle means that the process is considered, and a cross means that it is not considered.