# Peer review of "Global simulation of dissolved 231Pa and 230Th in the ocean and the sedimentary 231Pa/230Th ratios with the ocean general circulation model COCO ver4.0"

_Geoscientific Model Development, 2021_

## Author Comment (AC1)

**Response to Reviewer 1**

The authors greatly appreciate the reviewer for taking the time to this review. We will prepare a revised manuscript by taking into account all comments raised by the reviewer. Below, please find our responses (black) shown after the reviewer comments (blue). Thank you very much again for your review.

The authors present an offline simulation of dissolved and particulate Pa and Th, and sedimentary Pa/Th. The effect of bottom scavenging and scavenging efficiency depending on particle concentration are discussed through sensitivity experiments. The description of the modeling results are in great detail and the authors state that the model reproduces the observations reasonably well. However, there are still many places of model-data discrepancy. I understand that it is impossible to perfectly reproduce the observation, but the mismatch should at least be discussed. In addition, this study lacks novelty. It seems that it is confirming previous other modeling results and provides limited new results for improving our understanding of marine Pa and Th cycle. Also, a critical comparison of this modeling results of sedimentary Pa/Th with other modeling studies is missing.

We agree with the reviewer that there is room for improvement in our model. We will discuss the model-data discrepancy (raised by the reviewer's detailed comments) in more detail in the revised manuscript. However, our simulation which successfully reproduced the overall pattern of both the sedimentary Pa/Th ratio and dissolved Pa and Th of GEOTRACES data together with a detailed description and analysis about this model result is worth to be reported in GMD. On the other hand, the reviewer's comments about our insufficient comparison with previous studies are appreciated. Following the reviewer's comments, we will discuss the differences between our study and previous modeling studies (especially, by referring to more recent modeling studies than Siddall et al., 2005) in the revised manuscript.

Detailed comments are listed below:

In the introduction, the authors reviewed previous modeling studies. From the methods part, I see this study is offline modeling, which is a big difference from other modeling studies and should be emphasized.

Yes, our simulation is offline in that OGCM physical fields were calculated in advance and given as boundary conditions. We think that this is not a problem because the Pa/Th tracer distribution does not affect the physical fields at all (in other words, the simulated Pa/Th distribution does not depend on whether the model is offline or online).

Line 100, why not using observations for calcite and opal? Why calcite and opal export productions are calculated using ratio and POC observation?

There is no available observational global dataset for POC, calcite, and opal export flux. Therefore, we estimate them from chlorophyll concentrations and empirical rain ratios. The method is the same as that used in Siddall et al. (2005).

Line 154, sensitivity experiments in this study is using off-line model, it is not appropriate to call them "OGCM experiments", also later in the text.

As described above, the results of the Pa/Th simulation do not depend on whether the model is online or offline. Therefore, our simulation is equivalent to OGCM experiments.

What is the thickness of the nepheloid layer in the model?

Since we set the ocean deepest layer of each grid as the nepheloid layer, the thickness of the nepheloid layer depends on the thickness of the corresponding deepest ocean grid cell. The thickness increases with depth, from 5 m at the surface to 250 m at the deepest layer. This treatment is the same as Rempfer et al. (2017).

Rempfer et al., (2017) includes the bottom scavenging in their model, how this bottom scavenging implementation in this study differ from Rempfer et al., (2017)?

Our implementation is basically the same as Rempfer et al. (2017). More specifically, the concentration of lithogenic particles given in the deepest layer is the same as that used in Rempfer et al. (2017) (L.159). However, because the formulation of the partitioning coefficient (K) is not the same as Rempfer et al. (2017), the actual strength of bottom scavenging depends on its formulation and choice of their model parameters. Therefore, it was difficult for us to obtain appropriate parameters about K from Rempfer et al. (2017) and we needed to find appropriate parameters from our BTM-EXP (Fig.3). This point will be more clearly described in the revised manuscript.

Line 170, C_total and C_ref, are they global average or values in each grid? Is C_total on each grid and C_ref a global mean?

Yes, the value of C_total is differently specified on each grid whereas C_ref is given as a globally uniform. We will describe the meaning of the variables more precisely in the revised manuscript.

Figure 3, although with bottom scavenging, the Pa_d concentration decreases below ~3,000m, from the vertical profile, the observation shows similar values below 3,000m

with no decreasing trend with depth, but the model shows a clear decreasing trend (Figure 3d). How to explain this model-data discrepancy? It seems that the bottom scavenging is too strong.

The dissolved 231Pa is shown in Figure 2 (not Figure 3) and 230Th in Figure 3; therefore, it is not clear which the reviewer refers to. Here, we assume it is probably about 231Pa.

For dissolved 231Pa, introducing bottom scavenging helped to reproduce the concentrations seen in the data at depths below 3000m. However, as the reviewer pointed out, the model tends to simulate lower concentration than the observations below 3000m, which needs to be improved. The improvement was not possible simply by reducing the bottom scavenging (i.e., specifying the smaller K_bottom than Fig.2c/d), therefore more fundamental improvement appears to be required. One possibility is that our treatment of the nepheloid layer (i.e., the thickness of the ocean deepest layer) may be too simple and needs to be modified so that the thickness of the nepheloid layer is more realistically specified. This model-data discrepancy and its possible reason will be discussed in the revised manuscript.

Line 221, the authors pointed out the lower than observation Th_d, what could potentially cause this mismatch? Can it be improved in this model?

This point was a motivation of our PCE_EXP experiment and was already discussed in the manuscript. Please refer to the experiment introducing a dependence of scavenging efficiency on particle concentration (PCE_EXP), which improved this mismatch (Figure 5d).

Line 250-251 "Similar features are also found for dissolved 230Th". For Pa_d, the maximum around 3,000 is more or less reproduced. But for Th, the maximum in the model is much deeper than the observation. Near 30W, the simulated mid-depth Th_d is much lower than the observation. Also in Figure S6.

Thank you for pointing out this interesting feature. We are also curious about this difference between 230Th and 231Pa (i.e., successful simulation for 231Pa, but not for 230Th). The introduction of more realistic bottom scavenging and the consideration of the effects of particles from the continental shelf and hydrothermal vents may help to improve the model-data agreement for 230Th. We will discuss this point in the revised manuscript.

Line 251-252, the authors state that the particulate Pa and Th are well reproduced. It is not obvious in Figure S5c and d due to the colormap scale. In Figure S5c, observations are all in dark blue (hard to tell the value from the color bar), but simulation has some

green-yellow values. What processes in the model cause these high values in Pa_P? Also in Figure S5, Pa_P/Th_P should also be compared with observation.

Following the reviewer's advice, we will modify the figures to visualize them more clearly. As shown in the response to the comments above, the high concentration of particulate 231Pa may be related to the fact that, as with dissolved 231Pa, strong boundary scavenging is not considered in the model. We will also add figures of 231Pa/230Th ratios in the water columns in each GEOTRACES section.

Line 305, results about 1D_EXP. Sedimentary Pa/Th in the Atlantic is influenced by AMOC and particle flux effect in Siddall et al., (2005), probably AMOC is the first order factor (Gu & Liu, 2017). In this study, transport includes both of these effect. More experiments can be carried out to separate the effect of ocean currents and the diffusion caused by particle effects. In this off-line model, it is computationally achievable.

As we mentioned below (our response to your comment on Line 433-434), particle fields were not calculated in the model but specified as boundary conditions in our approach. Therefore, the particle fields are unchanged (i.e., there is no particle flux effect) in all our experiments. We feel that the reviewer misunderstood this point.

How is simulated sedimentary Pa/Th in CTRL and 3D_EXP compare with observation quantitatively? In Rempfer et al., (2017), the bottom scavenging is suggested to not affect Pa_P/Th_P to a small extent and also not affect the relationship between Pa_P/Th_P and AMOC. Does this study support the results in Rempfer et al., (2017)?

We will examine the difference in the quantitative agreement of sedimentary 231Pa/230Th ratios between CTRL and 3D and add it to Table S1. For addressing the relationship to AMOC, the experiment for changing the strength of the AMOC is required, which is beyond the scope of the manuscript.

Also, how is the sedimentary Pa/Th in this study compare quantitatively with other previous models? This is important because sedimentary Pa/Th is an important paleo proxy.

We will summarize the model-data agreement of sedimentary 231Pa/230Th ratios in Table S1. Also, we will discuss a comparison of the agreement with other modeling studies.

Line 334-336 is confusing. Please rephrase to make it more clear.

This sentence means that most of these radionuclides are present in the dissolved phase, so the change in the ocean transport of the dissolved (not particle) phase caused the change in the distribution of the particle phases. We will modify the sentences more clearly.

Line 339-340, what is the "ocean transport" here mean? Southward transport by the lower limb of Atlantic Meridional Overturning Circulation or diffusion?

The "ocean transport" here means the total difference between 3D_EXP and 1D_EXP (see Table 3) including advection and diffusion. As the reviewer mentioned, we think that the southward transport by the lower limb of AMOC is one of the main processes controlling the "ocean transport" effect.

Line 340-342, "At the same time…; as a result…" I cannot follow the logic here.

This statement is related to Line 344-366. Oceanic transport decreases dissolved 231Pa in the low latitudes and increases dissolved 231Pa in the Southern Ocean. As the increased dissolved 231Pa is adsorbed by particles in the Southern Ocean, particulate 231Pa increases (Figure 7a, 7c). The statement will be modified to "As described above, this ocean transport effect is also acting to particulate 231Pa via changes in dissolved 231Pa; as a result…".

Line 379: Residence time is calculated in CTRL_EXP and Siddall_EXP, with bottom scavenging the residence time is significantly decreased. However, the residence time in CTRL_EXP is similar to the residence time in (Gu & Liu, 2017) which does not include bottom scavenging. Does this mean the correct residence time does not necessarily need bottom scavenging?

Yes, we think that residence time depends on "total" scavenging efficiency in the ocean. Although the incorporation of bottom scavenging leads to an increase in scavenging efficiency and tends to reduce the residence time, bottom scavenging is not the sole process that controls the residence time. Therefore, for example, the model which specified the relatively stronger affinity to the particle can lead to shorter residence time even if the model does not include the bottom scavenging.

Line 426 "Part of the error comes from the oceanic flow fields simulated in the ocean model". How is the oceanic flow simulated in this model? It can be verified against products such as ECCO (Fukumori et al., 2018). Since Atlantic sedimentary Pa_P/Th_P is greatly influenced by AMOC, what does AMOC in this model look like? This information should be provided in the manuscript.

As described in section 2.1 of our manuscript, our physical flow fields were taken from the output of a pre-industrial simulation with MIROC and its details can be obtained in Kobayashi et al. 2015 and Oka et al. 2012. For the reader's information, we will consider showing a new Figure of the meridional circulation used in this study and information about its volume transport in the revised manuscript.

Line 433-434, why not in this study? Results presented in this study are mostly confirmation of previous studies. With the efficiency of this off-line model, more things can be done for example test this particle fields effect on Pa and Th.

Please note that particle fields were not calculated in the model but specified as boundary conditions in our approach. The specified distribution of biological particles is estimated by satellite-based observations. Since the bias of the particle field affects the distribution of 231Pa and 230Th, our approach has advantages over the other studies where the particles are explicitly simulated in the model. For the particle field, we use the most realistic distribution, so we do not conduct experiments to change it in our study.

The summary is too verbose. Line 435-480 is repeating things in the previous section.

We will revise the Summary section to avoid verbose statements.

Line 486-489 simulated sedimentary Pa/Th under glacial times are also discussed in a 2-D model (Lippold et al., 2012) and recently in a 3-D model (Gu et al., 2020).

We will refer to previous glacial modeling studies by citing the references that the reviewer pointed out.

Quantitative model data agreement (Pa_D, Th_D, Pa_P, TH_P, and sedimentary Pa/Th) and also residence time can be summarized in a table for different experiments. In that way, the performance of different experiments can be clearly seen.

We will show the model-data agreement and the residence time of each variable in Table S2 in the revised manuscript.

Minor issues:

Colorbars only have the highest and lowest values, it is not easy to tell the values in the middle.

We will modify the color bar to make its values easier to read.

Line 38, particulate organic carbon.

Thank you. We will change the word.

Line 47, Gu & Liu, (2017) also show AMOC change on sedimentary Pa/Th. Also, in Gu & Liu, (2017), the particle change due to freshwater and its impact on sedimentary Pa/Th is examined and should be cited in line 432.

We will cite Gu & Liu (2017) also in line 432.

Figure 6e, a lot of observations overlapped. It is hard to tell the color.

We will try to modify the figure to make it easier to read.

Figure 4 can be plotted in one figure as Figure 4 in Rempfer et al., (2017). In this way, the relative difference between different experiments is clearly shown.

Thank you for this suggestion. However, because we focus on two parameters (i.e., $K_{ref}$ and $K_{bottom}$), we think that dependency on these two parameters is easy to read in our Figure 4 rather than combing them all in one figure.

---

## Author Comment (AC2)

**Response to Reviewer 2**

The authors greatly appreciate the reviewer for taking the time to this review. We will prepare a revised manuscript by taking into account all comments raised by the reviewer. Below, please find our responses (black) shown after the reviewer comments (blue).

This study explores the processes that control the distribution of 231Pa and 230Th in the oceans and underlying sediments using COCO V4.0, an Ocean General Circulation Model (OGCM), from Hasumi 2006.

They implemented 231Pa and 230Th in the model using offline tracer simulations based on physical fields from COCO. They implemented bottom scavenging as well as a "dependence of scavenging efficiency on particle concentration" in the model.

General comments

- The most puzzling aspect of this manuscript is the lack of use of recent modeling results and the almost total lack of comparison with these model simulations (e.g. Missiaen et al., 2020a and 2020b; van Hulten et al., 2018; Rempfer et al., 2017; Lippold et al., 2012; Luo et al., 2010; Dutay et al., 2009; Roy-Barman, 2009). It is all the more surprising that most of these papers are cited by the authors although mostly as examples of recent publications instead of being analyzed in depth and compared to the COCO model outputs. A more thorough assessment of these new model simulations and how / why they agree / differ from the simulations presented in the manuscript must be done before publication.

The authors are appreciated for the reviewer's comments. In the original manuscript, we intended to provide a closed-form description of our model results, but we agreed with the reviewer that comparisons with the other modeling studies were also important and needed to be included in the revised manuscript. Therefore, we will cite the recent studies presented not only in the Introduction section but also in other sections and compare our results with them in the revised manuscript.

- Similarly, the choice of comparing the COCO model outputs with those of one of the earliest models used for 231Pa and 230Th, namely the model from the Siddall et al., 2005 study, is very disappointing as it misses out all the improvements made by the newer modeling studies and most of the conclusions drawn from these more recent simulations, most of which representing a significant improvement from the Siddall et al. (2005) model. The authors need to carefully and thoroughly justify their choice. Nevertheless, an in-depth discussion to compare their model outputs and conclusions with that of the more recent modeling studies is needed and should not be limited, as it

is the case in the present manuscript, to a comparison with the Siddall at al. (2005) simulation.

As mentioned above, we will compare our results with not only Siddall et al. (2005) but also other recent modeling studies in the revised manuscript. We think that our choice of reference to Siddall et al. (2005) is useful for demonstrating what kinds of new model treatment/parameters are required from this classic model for reproducing the recent observations from the GEOTRACES database. Our improvement comes from (1) incorporation of bottom scavenging, (2) choice of larger partitioning coefficient for 230 Th (Kref_230Th), and (3) inclusion of particle concentration effects to Kref_230Th. The first point (i.e., bottom scavenging) was already discussed in the previous study (Rempfer et al., 2017); our model confirmed its importance, and this point itself is not new. However, we showed that the performance of 230Th modeling is not enough to be improved simply by introducing the bottom scavenging and points (2) and (3) are required for its improvement. As the reviewers pointed out, the other recent modeling studies also showed improvement from Siddall et al. (2005) but are not necessarily the same way as the direction of improvement of our model results (e.g., Dutay et al. 2009 and van Hulten et al., 2018 focused on consideration of different particle size). We also emphasize here that our simulation of 231Pa/230Th is based on the ocean general circulation model (which is not a simplified model such as a 2D model or reduced complexity model). In the revised manuscript, we will discuss our model results by adding a comparison of our results with recent modeling studies.

- In the same vein, there is a great lack of recent literature analysis on 231Pa and 230Th, e.g. the recent review by Costa et al. (2020) or the recent findings of Missiaen et al. (2018) on the effect of the detrital (238U/232Th) activity ratio on the calculation of 231Pa and 230Thxs are neither discussed or cited. A lot of the effects that the authors are discussing in their manuscript is actually discussed in details for 230Th in the review paper by Costa et al. (2020).

Thank you for providing the references. We know that Costa et al. (2020) is a very nice review paper about "230Th normalization". 230Th normalization (which is a tool for reconstruction the sediment flux) is not a topic of our study but we found that this paper also includes some helpful information on 230Th modeling (section 5). We also thank you for providing paper information on Missiaen et al. (2018) about recent finding on the influence of lithogenic and authigenic 230Th on 230Th in sediments. These literatures will be cited in the revised manuscript.

- The literature used to discuss the effect of particles type and distribution is neither the first/pioneering papers on the topics nor the latest. The authors should read the review by Costa et al. (2020) and look at the modeling results of Missiaen et al. (2020b) and references therein. These results should be both mentioned in the state-of-the-art section

We will describe the influence of the particle field on sedimentary 231Pa/230Th ratios with appropriate citations of previous studies including Missiaen et al. (2020, QSR) which discussed that changes in the particle can affect 231Pa/230Th ratios.

- Similarly, the older literature is fundamentally overlooked. The term "boundary scavenging" has been defined and used by Anderson et al. (1983b). Part of what the authors seem to define as a discovery on the effect of particle concentration on scavenging is actually perfectly defined and modeled by Anderson and co-authors is this paper and subsequent papers. This leads to a conceptual problem L356-369 (see also comment on L194 below).

In our understanding, although this paper does not use the term "boundary scavenging" (they used "intensified scavenging" or "near-bottom scavenging"), the concept of "boundary scavenging" was actually introduced in this paper as the reviewer pointed out. We will cite Anderson et al. (1983) "Removal of 230Th and 231Pa at ocean margins" as a pioneering study about boundary scavenging in the revised manuscript.

- Several sentences or model presentation are very vague, e.g. in equation 4a, there is a term "Transport" (L116-120) defined as representing transport by advection, diffusion and convection. These are 3 very distinct physical processes in their formulation, why is the term "Transport" not explicitly given? What does the term "convection" represent in the oceans. There is no bottom heating so I have great troubles understanding what the authors mean here.

We think that the equation 4a is a very standard expression for representing ocean tracer concentration (e.g., this is equivalent to equation 9 of Siddall et al. 2005). As the reviewer pointed out, the transport term includes oceanic advection, diffusion, and convection. The convection term is represented in the model by so-called "convective adjustment" (e.g., Yin and Sarachik, 1994) where unstable stratification leads to very large vertical mixing. The notation of "transport" is also used in previous similar modeling studies (Rempfer et al., 2017; Gu and Liu, 2017; Missiaen et al., 2020, CP).

- I am very puzzled by the use of equation (10) (L169) for both 230Th and 231Pa. The partition coefficient cannot be the same for both radionuclides as they as have different behaviors. The value of the exponent used here (-0.42) has been given by Henderson et al. (1999) for 230Th and is indeed not valid for 231Pa. I do not see what can be achieved by using the same reference partition coefficient for both isotopes.

There is a misunderstanding in this reviewer's comment. We introduced the dependence of particle concentration only for 230Th (not for 231Pa).

- L90: there is one class of settling velocity in the model presented here. There are two classes in van Hulten et al. (2018). Since the authors discuss the effect of the concentration of particles on scavenging, they should discuss the effect of having one vs. more classes of settling speed on their conclusions

van Hulten et al. (2018) introduced multiple size classes of particles, and we understand that specifying the different settling velocity depending on size classes is one of the important aspects in their study. On the other hand, specifying the different settling speeds is unavailable in the framework of our scavenging model and will require a major upgrade of its model formulation. Therefore, its direct evaluation is difficult in our model. However, in the revised manuscript, we will discuss the effect of specifying settling speed, for example, by comparing our results with previous studies introducing this effect such as van Hulten et al. (2018) and Dutay et al. (2009).

- L194: The authors say they included bottom scavenging in benthic nepheloid layers. This is a very important aspect of the model. However, how this is done is not explained. More explanations of this very important aspect are necessary, especially considering the objective of the journal.

We have already described bottom scavenging in the section of experimental design (see L156-162) but will explain more clearly how bottom scavenging was introduced in the revised manuscript.

- Amongst the conclusions, some are included in the equation. The fact that 231Pa is more affected by advection is 1) the basis for using Pa/Th as a proxy for ocean circulation and has already been verified by several models, and 2) is somehow imbedded in the equations of scavenging.

Although the conclusion that the advection of 231Pa is the most important for sedimentary 231Pa/230Th ratios is the same as in previous studies, it is notable that the contributions of the transport and bottom scavenging of each element are evaluated separately. We also believe that our successful more result about both 231Pa and 230Th along with GEOTRACES sections is also worth to be reported in GMD.

- English should be proofread. The meaning of several sentences remains very ambiguous or unclear.

The manuscript was already checked by a professional English proofreading service, but the revised manuscript will also go through English editing again.

To conclude on these general comments: the model and its interpretations seems

detached from what is already known on Pa/Th both in the water column and the sediment from both modeling and data studies. This manuscript shows a lack of thorough reading (state-of-the-art) of the most recent (last 10 years) literature on the subject and lacks discussion of these recent findings / conclusions. The choice of using one of the oldest model to compare these new simulation results is very odd and thus lacks a great part of the novelty added by more recent studies. There are also several conceptual problems that need to be addressed.

As mentioned above, the other recent modeling studies also showed improvement from Siddall et al. (2005) but are not necessarily the same way as the direction of improvement of our model results. We also emphasize here that our simulation of 231Pa/230Th is based on the ocean general circulation model (which is not a simplified model such as a 2D model or reduced complexity model). Our simulation which successfully reproduced the overall pattern of both the sedimentary Pa/Th ratio and dissolved Pa and Th of GEOTRACES data together with a detailed description and analysis about this model result is worth to be reported in GMD.

Following the reviewer's comments, in the revised manuscript, we will appropriately cite the recent findings of models and observations. We will then provide a more detailed discussion of the comparison between our work and recent modeling studies.

Specific comments

- L24-25: if one wants to cover the all date range, there are more recent papers than Bohm et al. (2015), e.g. Sufke et al., 2020 or Waelbroeck et al. 2018

We will add the suggested literature.

- L30: there is also Henderson and Anderson 2003 review that gives a large range of residence times (see also Costa et al., 2020 for 230Th)

We will add the information of residence time from the suggested literature (130 years for 231Pa and 20 years for 230Th in Henderson and Anderson 2003).

- L36: for the LGM/Holocene comparison, there are more appropriate references, such as Lippold et al., 2014 which is a modeling and compilation of Atlantic data for the LGM vs. Holocene.

We will appropriately cite the suggested previous studies focusing on the LGM.

- L44 and after: several references missing or not cited appropriately. Many of the references cited cover several aspects of the Pa/Th modeling rather than only a specific aspect as the citation format made by the authors suggests.

We will cite and discuss literature covering all aspects of 231Pa/230Th modeling.

- L64: GEOTRACE database: cite

We will add a reference to the GEOTRACES project.

- L76: 43 vertical layers: are these of uniform or different heights. Be more precise.

We will add a more detailed description of our model grid.

- L81: how do you assess that you reached a steady state? explain

We will specify the criteria for determining the steady state.

- L81: Explain why you choose 100 years average rather than another number

In fact, since the residence time of 231Pa and 230Th is at most a few hundred years, they reach a steady state in about a thousand years, and almost no change in the average concentration of the entire ocean occurs. To remove short-term fluctuations and analyze the ocean mean state, the model is integrated over 3,000 years and the last 100-year average is used in the analysis.

- L171: "reference concentration". It is very unclear to me, based on the information given here what is the reference concentration. More details should be given.

This reference concentration is the standard concentration for introducing the dependence of scavenging efficiency on particle concentration.

---

## Referee Report (RR1)

The authors nicely improved the manuscript in this revised version. They have answered most of my comments / questions / suggestions / concerns. I still have a few comments that needs to be answered and a few minor adjustments are needed before publication.

One of my concern is that the justification for choosing a comparison with the simulation(s) made by Siddall et al. (2005), i.e. rather old results, which have since been discussed and improved, does not clearly appear. The study will indeed gain from a justification of this choice that should be given at the beginning of the manuscript, i.e. end of the "Introduction" and/or "Experimental design".

As underlined by the other reviewer, the fact that the model is offline is not correctly emphasized. From my point of view, it is not a weakness but a strength from this model because it makes it indeed easier to manipulate.

Figures with "vertical profile averaged horizontally" (1, 2, 3, 4 and 5): I am not sure I understand this term correctly. An explanation must be provided. Does it mean you choose 1) to make an average of all the dissPa concentrations at a given depth along the entire transect? Or 2) to average the value along a given latitude for the entire basin for each layer?

Then what is the representativeness of this averaged value (orange points) in the Atlantic because: case 1) The north and south Atlantic have very different behaviors with 231Pa being strongly entrained in the AMOC In the north and 231Pa being strongly scavenged by opal-rich particles in the south. Case 2) the west and east basins have different behaviors for diss Pa (diss Th probably to a lesser extend).

L38-41. As you described the conclusions based on the data analyses (lines above), you should also describe the main conclusions of the 2 models you are citing in this subsection.

L45. "sinking particles scavenge 230Th more strongly". I suggest to replace strongly by efficiently.

L45-47. While the work by Chase et al. is highly cited, there were much earlier studies showing the effect of opal such as Rutgers van der Loeff and Berger (Deep Sea Res. 1, 40, 1993) or Walter et al. (Earth Planet Sci Lett 149, 1997). Some of these studies are cited later in the manuscript but should also appear in the Introduction as this effect has been known for a long time.

L48-51. Citations should be proofread, e.g. Rempfer et al., van Hulten et al., Missiaen et al., 2020a are not 2D ocean models: please carefully check these references and to which model they correspond.

L63-65. While the approach was different from that of the authors' model, the effect of efficiency of scavenging depending on particle concentration has been explored by recent models, e.g. van Hulten et al. (2018); Missiaen et al. (2020a and 2020b), even if the approach was different. It should be mentioned here.

L124. I suggest that, for non-model specialists, you explain the term "convection" as you did in your answer to my initial review. This was a clear and short explanation that could make your paper more accessible to a broader audience.

L165. This part needs clarification.

You write that the benthic nepheloid layer is 50 to 130m above the bottom. Do you mean thickness? i.e. bottom to bottom+50m for 50m?

In addition, in your answer to reviewers, you say that the thickness of the nepheloid layer increases from 5 to 250m. There is a discrepancy with what is written in the manuscript. Please clarify this point.

L250-253. Since you are comparing the results of your simulations to that of other models for the Southern Ocean when discussing particulate Pa and Th (section 3.3) it would be good that have a similar comparison for the dissolved Pa and Th at the end of section 3.2. even if the other models indeed use slightly different approaches to simulate the effect of changing adsorption coefficient in the Southern Ocean (e.g. Rempfer et al., 2017 and Missiaen et al., 2020a).

L301. typo: Missiaen (not Messiaen)

L320. typo: GEOTRACES (not GAOTRACES)

L334-336. "In this transect, the observational data shows a clear signal associated with hydrothermal vents": please explain the underlying mechanism, i.e. how would hydrothermal vents affect both diss. and part. Pa and Th concentrations. A reference is also needed. May be also the earlier findings of this mechanism, e.g. Shimmield and Price, Geochim Cosmochim Acta 52, 1988.

L337. replace "our scope" by "the scope of this study"

L341. "we discuss about": remove "about"

L401. "as a matter of course" replace by "as a matter of fact"

L405. "231Pa transported toward the Southern Ocean is expected to be immediately removed there due to the high opal flux". This is an overstatement. Immediately should be replaced by "quickly" or "very quickly" or "quicker than in the open ocean". Both data and modeling studies show that there is still some Pa exported within the Southern Ocean

L408-410. "This result implies that scavenging of 230Th is not so efficient in the Southern Ocean as previously expected due to the dependence of scavenging efficiency on particle concentration.". In Missiaen et al. (2020a), they simulate the effect of halving of 1) the total particle flux, 2) the POC, 3) the CaCO3 and 4) opal. They show that for Th, it results in increasing the dissolved 230Th concentration in the Southern Ocean in case 1) and further show that the main effect comes from POC and opal. How does it compare with your results on scavenging efficiency and particle concentration? Can you link both results?

L453: also add the ref to Henderson and Anderson for residence time

L498. "(i.e., specifying the smaller  $K^{\text{Pa}_{\text{bottom}}}$  than Figs. 2c and 2d)" do you actually mean: "(i.e., specifying smaller  $K^{\text{Pa}_{\text{bottom}}}$  than in Figs. 2c and 2d)"?

L819. "Mangini" not "Mangianini"

---

## Author Response (AR2)

**Response to Reviewer 1**

The authors are greatly indebted to the reviewers who took the time to write this review. We revised manuscript by considering all comments raised by the reviewer. Below, please find our responses (black) shown after the reviewer's comments (blue).

I appreciate the authors' efforts in revising the manuscript by adding a section of the comparison with other modelling studies. However, the comparison is too descriptive without quantitative assessment and in-depth discussion. The revised manuscript still lacks "novelty" and new insights into the marine Pa and Th cycle. As the authors state in the text, bottom scavenging and dependence of scavenging efficiency on particle concentration are all confirming previous studies. Limited new insights are provided in the current manuscript.

Because our manuscript is submitted as a model description paper, we believe that it is worth to report our effort about simulating the global distribution of 231Pa and 230Th in seawater and sediment by using our 231Pa/230Th model with OGCM COCO ver4.0. Although the individual processes are already reported in previous studies as pointed out by the reviewer, it is valuable to demonstrate that combination of these processes can reproduce the overall structure of observed Pa and Th distribution from our model. Considering the reviewer's comment and the editor's suggestion, we changed the title of the manuscript in this revision: old tile "An investigation into the processes controlling the global distribution of dissolved 231Pa and 230Th in the ocean and the sedimentary 231Pa/230Th ratios by using an ocean general circulation model COCO ver4.0" $\rightarrow$  new title "The global simulation of dissolved 231Pa and 230Th in the ocean and the sedimentary 231Pa/230Th ratios by using an ocean general circulation model COCO ver4.0".

**Major:**

1. "Novelty". With a computational efficient offline model, more ideas can be tested which will help the future improvement of Pa and Th modelling in those 3-D online models. For example, in section 4.5, the thickness of the nepheloid layer is proposed. Why not carry out some sensitivity experiments to see how the nepheloid layer thickness affect Pa and Th? Also, the other reviewer suggested one vs more classes of settling velocity, which is also something "new" to test. But the authors' response is "specifying the different settling speeds is unavailable in the framework of our scavenging model and require a major upgrade of its model formulation", which is not acceptable. The authors should take the advantage of the offline framework and think carefully about experimental design to really advance our understanding of marine Pa and Th cycle.

As described above, main aim of our manuscript is to describe our 231Pa and 230Th model and report detail results of our simulation. Following the reviewers' previous comments, description about the comparison with previous modeling results (section 4.1) were also added in the previous revision. Therefore, we believe that the content of our present manuscript meets the objectives and standards of the GMD journal. As for the reviewer's comment about the thickness of nepheloid layer, we set it as the thickness of the deepest grid of each ocean grid as in Rempfer et al. (2017). Therefore, it is not possible now to freely change its thickness (also see our reply to your comment 2). As for the comment about the settling velocity, its dependency was already reported in Siddall et al. (2005); the choice of settling velocity had negligible impact on the sediment 231Pa / 230Th activity ratios (see Fig.6 in Siddall et al. 2005). Also note that our and Siddall's model assumed that the settling velocity is the same between particles, and because of this assumption, specifying the different value depending on particle type is not possible in our model framework. Therefore, as for suggestion about one vs more classes of settling velocity, we need to repeat our previous reply "specifying the different settling speeds is unavailable in the framework of our scavenging model and require a major upgrade of its model formulation". Finally, as for the reviewer's comment about "offline" model, please note that our model is not necessarily computationally efficient in that 3D tracer calculation needs to be explicitly conduced in our model, although the tracer calculation is separately performed from the physical fields. We added the following explanation in "Materials and Methods" section of the revised manuscript.

"The "offline" means that calculation of tracer is separately performed from that of physical field; since the distributions of 231Pa and 230Th do not affect the physical fields at all, the results do not depend on whether the model is "offline" or "online". The offline tracer model makes it easier to perform various sensitivity experiments."

2. How nepheloid layers are simulated is not described in detail. Although line 161-165 describes a little bit, it is not explicit enough for others to follow and reproduce.

To make the methodology clearer, we explained how to introduce the bottom scavenging as

follows in the revised manuscript.

"Second, we perform an experiment named BTM\_EXP, in which we additionally take bottom scavenging into account. Following Rempfer et al. (2017), we simply set the deepest model grid layer as the nepheloid layer. The thickness of the nepheloid layer becomes equal to the thickness of the corresponding the deepest model grid layer which varies between 5 and 250 m depending on the depth. The intensity of the bottom scavenging depends on two parameters: the partition coefficient ( $K_{bottom}$ ) and the concentration ( $C_{bottom}$ ) of the bottom particles. Our treatment about  $C_{bottom}$  is the same that in Rempfer et al. (2017); we assume a globally uniform value for  $C_{bottom}$  (6.0×10-8 g cm-3) which is within the range of  $4.0 \times 10^{-8}$  to  $1.65 \times 10^{-6}$  g cm-3 observed in the benthic nepheloid layers in the North Atlantic (Lam et al., 2015). As for  $K_{bottom}$ , because our formulation of the reversible scavenging is not the same as Rempfer et al. (2017), we needed to find its appropriate parameter value. For this purpose, we perform a number of simulations with different bottom scavenging intensities by changing the value of  $K_{bottom}$ ."

3. Section 4.1: I appreciate the authors adding this section to have a comparison with other modelling works. However, the majority of this session is what we already know: without bottom/boundary scavenging, the model will overestimate deep dissolved Pa and Th concentrations, which is already pointed out/discussed in previous literature. This revision still lacks in-depth and quantitative comparison with other modelling works, for example, Rempfer et al., 2017.

Considering the reviewer's comment, we modified the section 4.1 to emphasize the comparison with the previous studies including Rempfer et al. (2017). The discussion about comparison with previous 231Pa/230Th modeling studies on model-data comparison, which was previously stated in the second half of the section 4.4, was moved to the section 4.1 and we tried to make quantitative comparison there in the revised manuscript.

4. Section 4.3: Three experiments (1D, 3D, CTRL) are used to decompose the processes controlling sedimentary Pa/Th. In 1D, with only reversible scavenging, the sedimentary Pa/Th is 0.093, which is common knowledge and the dissolve phase distribution in this scenario has already been discussed in previous literature (e.g., Siddall et al., 2005). Line 358-Line 367 seems to be redundant and repeats what we already know. Similar for Line371-372.

As you mentioned, these statements are not new findings. In these statements, we just intended

to confirm that our result can be interpreted from common knowledge. We still think that these statements are helpful to understand/discuss the differences between our three experiments. In the revised manuscript, we added the words "although it is well known from previous studies," in this discussion.

5. Section 4.3: The "new" finding in this section to me is how ocean transport and bottom scavenging on each Pa and Th change sedimentary Pa/Th. This part can be improved by quantitatively comparing the effect on Pa and effect on Th; and also compare with Rempfer et al., 2017 results, which has the experiment with ocean circulation (similar to 3D) and ocean circulation & bottom scavenging (similar CTRL here), so that the robustness of the current results (Figure 8e, 9) can be verified.

Thank you for this suggestion. As for your suggestion about comparing the effect on Pa and effect on Th, this was already discussed with Figure 9c (Pa bottom scavenging) and Figure 9d (Th bottom scavenging) in our manuscript. As for comparison with Rempfer et al. (2017), because they did not show the individual effect from Pa and Th, comparison of our Figures 9c and 9d with Rempfer et al. (2017) was not possible. It is also difficult for us to directly compare our Figure 8e with Rempfer et al. (2017) because differences in sedimentary Pa/Th distribution between the experiments are not explicitly shown in Rempfer et al. (2017).

**6. Figures 6f, 7f, 8f are mentioned in the text, but missing in the figures.**

We are sorry for the missing of the figures in our previous submission. We included these figures in the revised manuscript.

7. Line 450-452 "Our CTRL\_EXP..." The authors claim that their results are more realistic than others, based on their more realistic dissolved Pa and Th. This is not convincing. If the authors can provide the quantitative comparison (different model vs same observation, RMSD and correlation as Table S1) showing better model-data agreement, then it is convincing.

This statement refers to the comparison between the work of Siddall et al. (2005) and ours. It was rewritten as follows:

"Compared with Siddall\_EXP based on Siddall et al. (2005), our CTRL\_EXP can realistically simulate not only oceanic distribution of 231Pa and 230Th but also their residence time by

*introducing the bottom scavenging and the dependence of scavenging efficiency on particulate concentration.*"

Minor:

Line 25: There are many more references using Pa/Th sedimentary ratio on past ocean circulation, suggest adding "e.g.,"

We have revised the manuscript as you have pointed out.

Line 36-40: "For example..." and "Some modeling..." These two sentences are not logically coherent and it reads to me there is another sentence after the second sentence. I guess the authors intend to say the model suggests stronger or similar LGM AMOC, therefore implementing Pa and Th in the model is important?

Yes, to avoid the confusion, we modified the sentences as follows in the revised manuscript so that the detail description about the LGM AMOC is not explicitly mentioned.

"To use the sedimentary  ${}^{231}Pa/{}^{230}Th$  ratios as a proxy for ocean circulation in a more quantitative manner, the modeling about  ${}^{231}Pa$  and  ${}^{230}Th$  is important."

Line 49-50: Gu and Liu, 2017; Rempfer et al., 2017; van Hulten et al., 2018; Missiaen et al., 2020a are 3D ocean models but cited as 2D models.

Thank you for pointing this out. We have cited them as studies using 3D models.

Line 243-244: Authors state that PCE\_EXP matches better than KREF\_EXP. From Figuren5d, the agreement above 3,500m is obviously improved in PCE\_EXP, but below 3,500m, KREF\_EXP seems to agree better as the observation shows maximum value ~5km (also in KREF\_EXP), but in PCE\_EXP, the maximum value is ~4km. The better agreement with GEOTRACES in PCE\_EXP in Table S1 is probably contributed by the upper ocean. Why there is a difference between the upper ocean and the abyssal ocean. More details should be discussed.

The difference between PCE\_EXP and KREF\_EXP below 3.5km is explained by the change in

the partition coefficients (see Figure 5c). Due to introduction of the dependence of particle concentration on scavenging efficiency, the reference partition coefficient varies with ocean region in PCE\_EXP (Fig. 5c): in the deep ocean below 3,500m, it is significantly higher than  $1x10^{7}$  (=value assumed in KREF\_EXP) in the low-latitude regions. This caused the lower concentration below 3.5km in PCE\_EXP than KREF\_EXP. As for the underestimation of Th in the abyssal ocean, we added the following statement in the section 4.5 of the revised manuscript.

"The dissolved 230Th simulated in CTRL\_EXP (Fig. 5b) also tends to underestimate the observed concentration near the sea bottom. One possibility is that our treatment of the nepheloid layer (i.e., the thickness of the ocean deepest layer) may be too simple and needs to be modified so that the thickness of the nepheloid layer is more realistically specified."

**Line 371: "CTL\_EXP" typo.**

This is fixed in the revised manuscript. Thank you.

**Response to Reviewer 2**

The authors greatly appreciate the reviewer who took the time to this review. We prepared a revised manuscript by considering all comments raised by the reviewer. Below, please find our responses (black) shown after the reviewer comments (blue).

The authors nicely improved the manuscript in this revised version. They have answered most of my comments / questions / suggestions / concerns. I still have a few comments that needs to be answered and a few minor adjustments are needed before publication.

One of my concern is that the justification for choosing a comparison with the simulation(s) made by Siddall et al. (2005), i.e. rather old results, which have since been discussed and improved, does not clearly appear. The study will indeed gain from a justification of this choice that should be given at the beginning of the manuscript, i.e. end of the "Introduction" and/or "Experimental design".

We are grateful for the advice. Following the reviewer's comment, we added the following statement at the beginning of section 2.5 "Experimental design" in the revised manuscript.

"As stated in the Introduction, Siddall et al. (2005) was a pioneering 3D model for global simulation of both 231Pa and 230Th. This model is now a relatively old model and the reversible scavenging model introduced in this model is simpler than more recent models. However, this model appropriately reproduced the observed distribution of sedimentary 231Pa/230Th ratios as shown in their Fig. 2 which appears not necessarily inferior to that in more recent models. Therefore, in this study, we start with Siddall EXP where this most basic reversible scavenging model of Siddall et al. (2005) is introduced."

As underlined by the other reviewer, the fact that the model is offline is not correctly emphasized. From my point of view, it is not a weakness but a strength from this model because it makes it indeed easier to manipulate.

Thank you for this comment. Following your comment, we added the following statement in "Materials and Methods" section of the revised manuscript.

"The "offline" means that calculation of tracer is separately performed from that of physical

field; since the distributions of 231Pa and 230Th do not affect the physical fields at all, the results do not depend on whether the model is "offline" or "online". The offline tracer model makes it easier to perform various sensitivity experiments."

Figures with "vertical profile averaged horizontally" (1, 2, 3, 4 and 5): I am not sure I understand this term correctly. An explanation must be provided. Does it mean you choose 1) to make an average of all the dissPa concentrations at a given depth along the entire transect? Or 2) to average the value along a given latitude for the entire basin for each layer?

Then what is the representativeness of this averaged value (orange points) in the Atlantic because: case 1) The north and south Atlantic have very different behaviors with 231Pa being strongly entrained in the AMOC in the north and 231Pa being strongly scavenged by opal-rich particles in the south. Case 2) the west and east basins have different behaviors for diss Pa (diss Th probably to a lesser extend).

The figure caption was modified to be "vertical profile (the latitudinal mean along 30°W in the Atlantic Ocean)" in the revised manuscript. As you pointed out, this average masks the difference between north and south parts of the Atlantic basin. But this north-south difference is explicitly shown in other figures (e.g. Fig. 1a for Pa; Fig. 2c for Th) in our manuscript.

L38-41. As you described the conclusions based on the data analyses (lines above), you should also describe the main conclusions of the 2 models you are citing in this subsection.

By responding to the other reviewer's comment, this sentence was removed and replaced by the following statement in the revised manuscript.

"To use the sedimentary  $^{231}Pa/^{230}Th$  ratios as a proxy for ocean circulation in a more quantitative manner, the modeling about  $^{231}Pa$  and  $^{230}Th$  is important."

L45. "sinking particles scavenge 230Th more strongly ". I suggest to replace strongly by efficiently.

We have revised the manuscript as you have pointed out.

L45-47. While the work by Chase et al. is highly cited, there were much earlier studies showing

the effect of opal such as Rutgers van der Loeff and Berger (Deep Sea Res. 1, 40, 1993) or Walter et al. (Earth Planet Sci Lett 149, 1997). Some of these studies are cited later in the manuscript but should also appear in the Introduction as this effect has been known for a long time.

Thank you for the references. We have cited them in revised the manuscript.

L48-51. Citations should be proofread, e.g. Rempfer et al., van Hulten et al., Missiaen et al., 2020a are not 2D ocean models: please carefully check these references and to which model they correspond.

Thank you for pointing this out. We have cited them as studies using 3D models.

L63-65. While the approach was different from that of the authors' model, the effect of efficiency of scavenging depending on particle concentration has been explored by recent models, e.g., van Hulten et al. (2018); Missiaen et al. (2020a and 2020b), even if the approach was different. It should be mentioned here.

Thank you for the suggestion. We modified the sentence in the revised manuscript by citing the references the reviewer suggested.

"... this effect has not been directly considered by recent modeling studies but some studies have evaluated the impacts of changes in particle concentration and scavenging efficiency on the distribution of 231Pa and 230Th (van Hulten et al., 2018; Missiaen et al., 2020a and 2020b)"

L124. I suggest that, for non-model specialists, you explain the term "convection" as you did in your answer to my initial review. This was a clear and short explanation that could make your paper more accessible to a broader audience.

Because the "convection" term is included as a part of diffusion term in our offline tracer calculation, we simplified our explanation so that the term "convection" is not explicitly described in the revised manuscript.

L165. This part needs clarification.

You write that the benthic nepheloid layer is 50 to 130m above the bottom. Do you mean thickness? i.e., bottom to bottom+50m for 50m? In addition, in your answer to reviewers, you say that the thickness of the nepheloid layer increases from 5 to 250m. There is a discrepancy with what is written in the manuscript. Please clarify this point.

The values of 50-130 mean the depth above bottom of observed maximum suspended particulate matter reported in Lam et al. (2015, DSR). We set the deepest grid cell as the nepheloid layer. Therefore, the thickness of the nepheloid layer is equal to the thickness of the corresponding deepest grid cell. The thickness of grid cell increases with depth from 5 to 250 m in our ocean model. To make this point clearer, we modified the description as follows in the revised manuscript.

"Second, we perform an experiment named BTM\_EXP, in which we additionally take bottom scavenging into account. Following Rempfer et al. (2017), we simply set the deepest model grid layer as the nepheloid layer. The thickness of the nepheloid layer becomes equal to the thickness of the corresponding the deepest model grid layer which varies between 5 and 250 m depending on the depth. The intensity of the bottom scavenging depends on two parameters: the partition coefficient ( $K_{bottom}$ ) and the concentration ( $C_{bottom}$ ) of the bottom particles. Our treatment about  $C_{bottom}$  is the same that in Rempfer et al. (2017); we assume a globally uniform value for  $C_{bottom}$  ( $6.0 \times 10^{-8}$  g cm-3) which is within the range of  $4.0 \times 10^{-8}$  to  $1.65 \times 10^{-6}$  g cm-3 observed in the benthic nepheloid layers in the North Atlantic (Lam et al., 2015). As for  $K_{bottom}$ , because our formulation of the reversible scavenging is not the same as Rempfer et al. (2017), we needed to find its appropriate parameter value. For this purpose, we perform a number of simulations with different bottom scavenging intensities by changing the value of  $K_{bottom}$ ."

L250-253. Since you are comparing the results of your simulations to that of other models for the Southern Ocean when discussing particulate Pa and Th (section 3.3) it would be good that have a similar comparison for the dissolved Pa and Th at the end of section 3.2. even if the other models indeed use slightly different approaches to simulate the effect of changing adsorption coefficient in the Southern Ocean (e.g. Rempfer et al., 2017 and Missiaen et al., 2020a).

Thank you for the suggestion. We added the following sentences near the end of section 3.2.

"The distributions of 230Th simulated in previous modeling studies (e.g., Figs. 4 and 5 in Dutay et al., 2009; Fig. 2 in Siddall et al., 2005; Fig. 2 in Gu and Liu; Fig. 3 in Rempfer et al., 2017;

Fig. 12 in van Hulten et al., 2018; Fig. S3 in Missiaen et al., 2020a) are basically similar to our result (Fig. 6b); however, our simulation is the best at reproducing the high concentration in the Southern Ocean."

**L301. typo: Missiaen (not Messiaen)**

This is fixed in the revised the manuscript. Thank you.

**L320. typo: GEOTRACES (not GAOTRACES)**

This is fixed in the revised the manuscript. Thank you.

L334-336. "In this transect, the observational data shows a clear signal associated with hydrothermal vents": please explain the underlying mechanism, i.e. how would hydrothermal vents affect both diss. and part. Pa and Th concentrations. A reference is also needed. May be also the earlier findings of this mechanism, e.g. Shimmield and Price, Geochim Cosmochim Acta 52, 1988.

Thank you for the information. We explained the underlying mechanism related to hydrothermal activities in this transect as follows:

"It has been pointed out that trace metals from hydrothermal activities may cause additional removal of 231Pa and 230Th (Shimmield and Price 1988; Lopez et al., 2015; Rutgers van der Loeff et al., 2016; German et al., 2016). Along the GEOTRACES GP16 section, 230Th and 231Pa have been found to decrease with increasing trace metals of iron and manganese supplied from hydrothermal vents (Pavia et al., 2018)."

**L337. replace "our scope" by "the scope of this study"**

We have revised the manuscript by following your suggestion.

**L341. "we discuss about": remove "about"**

We have revised the manuscript by following your suggestion.

**L401. "as a matter of course" replace by "as a matter of fact"**

We have revised the manuscript by following your suggestion.

L405. "231Pa transported toward the Southern Ocean is expected to be immediately removed there due to the high opal flux". This is an overstatement. Immediately should be replaced by "quickly" or "very quickly" or "quicker than in the open ocean". Both data and modeling studies show that there is still some Pa exported within the Southern Ocean

Thank you for pointing out this. We have revised the manuscript by following your suggestion.

L408-410. "This result implies that scavenging of 230Th is not so efficient in the Southern Ocean as previously expected due to the dependence of scavenging efficiency on particle concentration.". In Missiaen et al. (2020a), they simulate the effect of halving of 1) the total particle flux, 2) the POC, 3) the CaCO3 and 4) opal. They show that for Th, it results in increasing the dissolved 230Th concentration in the Southern Ocean in case 1) and further show that the main effect comes from POC and opal. How does it compare with your results on scavenging efficiency and particle concentration? Can you link both results?

Thank you for this discussion. Following your suggestion, we added the following discussion in the revised manuscript.

"Missiaen et al. (2020a) demonstrated that the dissolved 230Th concentration in the Southern Ocean will increase if the effect of particle scavenging is halved and that most of this effect come from POC and opal. This implies the scavenging of 230Th is controlled also by the opal in the Southern Ocean. Together with their and our results, quantification about scavenging of Th by opal in the Southern Ocean may be a key for more accurate understanding of 231Pa/230Th ratios in the global ocean."

**L453: also add the ref to Henderson and Anderson for residence time**

We have cited the reference in revised the manuscript.

L498. "(i.e., specifying the smaller *KK*Pabottom than Figs. 2c and 2d)" do you actually mean: "(i.e., specifying smaller *KK*Pabottom than in Figs. 2c and 2d)"?

Yes. We have revised the manuscript as you have pointed out.

**L819. "Mangini" not "Mangianini"**

This is fixed in the revised manuscript. Thank you.

---

## Author Response (AR3)

Dear Dr. Paul Halloran,

The authors are grateful to the editors and reviewers for their constructive comments on our manuscript. According to the editor's suggestion, we revised the manuscript with changing the title, correction of unclear text and typos, and other minor modifications. Below, please find our responses (black) shown after the topical editor's comments (blue).

Non-public comments to the Author:

For clarity I suggest making the following change to the title:
The global simulation of dissolved 231Pa and 230Th in the ocean and the sedimentary 231Pa/230Th ratios by using an ocean general circulation model COCO ver4.0
To
Global simulation of dissolved 231Pa and 230Th in the ocean and the sedimentary 231Pa/230Th ratios within the ocean general circulation model COCO ver4.0

Thank you for this suggestion. We changed the title as follows:
"Global simulation of dissolved 231Pa and 230Th in the ocean and the sedimentary 231Pa/230Th ratios with the ocean general circulation model COCO ver4.0"
This is almost the same as your suggested title (expcept within→with) and we hope this title is also fine.

I suggest changing:
"Therefore, our result should be viewed as a confirmation of these previous results in this meaning."
To
"Therefore, our result should be viewed as a confirmation of these previous results"

Thank you for this suggestion. We modified the sentence as you suggested.

Regarding the sentence:

"However, it is emphasized that this study provides a new estimate of this contribution…".
Please state what "this contribution" is referring to. Dependence of the scavenging efficiency on particle concentration?

This contribution refers to the contribution from bottom scavenging. To make the sentence clearer, we modified it as follows in the revised manuscript.

*"However, we emphasize that this study provides a new estimate of the contribution of bottom scavenging to the distribution of..."*

Please correct this sentence. It presently does not make sense:

"As another remaining problem, as pointed out in previous studies (Rempfer et al., 2017; Lerner et al., 2020), it is not easy to reproduce the distribution of particle phase of these two radioisotopes than the dissolved phase.."

Thank you for pointing it out. To make the sentence clearer, we modified this sentence as follows in the revised manuscript.

*"As pointed out in previous studies (Rempfer et al., 2017; Lerner et al., 2020), the distributions of particle phases of $^{231}Pa$ and $^{230}Th$ are diffucult to be reproduced in the model compared with the dissolved phases."*

We also made change about the text around this sentence in the revised manuscript.

I would also strongly suggest that you quote p-values where you present correlations.

In the general meaing, we agree with the editor's suggestion about the importance of a significance test about the correlation. However, in our case, the correlation coefficient was used for the aim of quantifying the model-data agreement rather than judging whether two parameters (i.e. model and data, in our case) are correlated or uncorrelated. Therefore, we feel that showing information about significant test (p-values) becomes somewhat misleading. In addition, not only the correlation but also other metrics such as "slope" and "RMSE" were used for quantifying the model-data agreement in our manuscript (e.g. Table S1); we are afraid that increasing the amount of information may make our analysis more difficult to understand. Therefore, we believe that showing the value of the correlation without p-values is enough and appropriate for our purpose.